# Closing the methane gap in US oil and natural gas production emissions inventories

Jeffrey S. Rutherford [1], Evan D. Sherwin [1], Arvind P. Ravikumar [2], Garvin A. Heath [3], Jacob Englander [4], Daniel Cooley [5], David Lyon [6], Mark Omara [6], Quinn Langfitt [4] & Adam R. Brandt [1]✉

Methane (CH$_4$) emissions from oil and natural gas (O&NG) systems are an important contributor to greenhouse gas emissions. In the United States, recent synthesis studies of field measurements of CH$_4$ emissions at different spatial scales are ~1.5–2× greater compared to official greenhouse gas inventory (GHGI) estimates, with the production-segment as the dominant contributor to this divergence. Based on an updated synthesis of measurements from component-level field studies, we develop a new inventory-based model for CH$_4$ emissions, for the production-segment only, that agrees within error with recent syntheses of site-level field studies and allows for isolation of equipment-level contributions. We find that unintentional emissions from liquid storage tanks and other equipment leaks are the largest contributors to divergence with the GHGI. If our proposed method were adopted in the United States and other jurisdictions, inventory estimates could better guide CH$_4$ mitigation policy priorities.

[1] Department of Energy Resources Engineering, Stanford University, Stanford, CA, USA. [2] Department of Systems Engineering, Harrisburg University of Science and Technology, Harrisburg, PA, USA. [3] Joint Institute for Strategic Energy Analysis (JISEA), National Renewable Energy Laboratory, Golden, CO, USA. [4] Industrial Strategies Division, California Air Resources Board, Sacramento, CA, USA. [5] Department of Statistics, Colorado State University, Ft. Collins, CO, USA. [6] Environmental Defense Fund, Austin, TX, USA. ✉email: abrandt@stanford.edu

Methane ($CH_4$) is the principal constituent of natural gas and is also a potent greenhouse gas (GHG)[1]. During production of oil and natural gas (O&NG), some processes are designed to vent $CH_4$ to the air, and $CH_4$ is also emitted unintentionally via leaks in the system. According to the official United States (US) GHG inventory, $CH_4$ from O&NG operations are estimated to contribute ~3% of national GHG emissions (with 100 year GWP = 25,[2]). At the international level the contribution is approximately 5% (based on estimates from[3] and[4]). However, the uncertainty in this estimate, data gaps, and inconsistency with alternative approaches suggested a need for further evidence[5–8]. To this end, significant research in the past decade has investigated $CH_4$ emissions from the O&NG system.

The US Environmental Protection Agency (EPA) estimates O&NG $CH_4$ emissions in an annual Greenhouse Gas Inventory (GHGI)[9]. The GHGI uses a data-rich, bottom-up approach to estimate national $CH_4$ emissions by scaling up $CH_4$ emissions measurements from activities like well completions and gas-handling components like valves or seals. However, a recurrent theme consistently found in the literature is that the GHGI underestimates total US O&NG $CH_4$ emissions compared to observed values[10]. Brandt et al.[11] summarize the literature, and observe that national-scale estimates from large-scale field studies exceed the GHGI by ~1.5 times. This difference is sometimes referred to as the top-down/bottom-up gap[11–17], based on the differences in approach between the GHGI and the conflicting studies. Top-down studies determine total emissions from multiple sites via measurements from aircraft, satellites, or weather stations (e.g.[14–16,18–20]).

Some recent studies have used a meso-scale site-level approach which measures $CH_4$ down-wind of facilities (e.g., well-pads) to estimate total emissions of an entire site or facility (e.g.[21–24]). A recent synthesis of site-level data by Alvarez et al.[13] finds agreement between site-level results and top-down results, with a best estimate of supply chain emissions (including all equipment from production to distribution) ~1.8 times that of the component-level GHGI[25] (up to ~2.1× in the production-segment). Based upon their validation with top-down studies and consistency with Brandt et al.[11] results (in terms of exceedance over GHGI values), we consider Alvarez et al. to be the most reliable estimate to date of US O&NG supply chain $CH_4$ emissions.

Most emissions sources in the GHGI are derived using bottom-up methods. The bottom-up approach estimates overall $CH_4$ emissions by combining counts of individual components (or activities) with emissions per component/activity (the emission factor). The bottom-up approach allows for representation of sources at a high resolution, with 67 and 45 separate sources for the O&NG production segments, respectively[25]. Because of this high resolution, the GHGI is useful for development of $CH_4$ mitigation policies. For example, the Obama administration's Climate Action Plan developed recommendations using the relative contribution of emissions sources in the GHGI[26]. Also, the bottom-up framework of the GHGI is recommended for reporting national emissions under the United Nations Framework Convention on Climate Change (UNFCCC[27]), under which participating countries report their inventory of GHG emissions.

Despite important advances in our understanding of $CH_4$ emissions from the O&NG sector, questions remain. First, why does the bottom-up EPA GHGI underestimate $CH_4$ emissions compared to both site-level and large-scale top-down studies? Second, is this underestimation due to an inherent problem with the bottom-up methods used in the GHGI? Previous studies have noted that many of the underlying data sources of the GHGI were published in the 1990s and may be outdated[11,28,29]. The site-level synthesis study of Alvarez et al.[13] suggested that the divergence is likely due to a systematic bias in the bottom-up methodology that

misses super-emitters, a finding supported by others (e.g.,[11,30]). Recent work suggests that top-down measurement campaigns are capturing systematically higher emissions during daytime hours from episodic events[31]. However, this may not be true at a national level, as it has been noted that the upward bias of top-down measurements was likely explained by unusually high liquids-unloadings in the Fayetteville shale[13]. Some have attempted to construct alternative inventories (e.g.,[13,32,33]), however these attempts have not taken full advantage of the robust set of component-level data now available.

In this work, our contributions are threefold. First, we construct a bottom-up, O&NG production-segment $CH_4$ emissions estimation tool based on the most comprehensive public database of component-level activity and emissions measurements yet assembled. Our analysis boundary is the O&NG production segment which includes all active, onshore well pads and tank batteries (excluding inactive and offshore wells) and ends prior to centralized gathering and processing facilities (Supplementary Fig. 1). We focus on the production segment given its significant emissions (~58% of total supply chain $CH_4$ emissions in Alvarez et al.[13]) and the large difference between site-level estimates and the GHGI[13] (~70% of difference between Alvarez et al.[13] and the GHGI, Supplementary Fig. 2). Our approach differs from the GHGI in that it applies a bootstrap resampling statistical approach to allow for inclusion of infrequent, large emitters, thus robustly addressing the issue of super-emitters. Second, we use this tool to produce an inventory of US O&NG production segment $CH_4$ emissions and compare this with the GHGI and previous site-level results. Here, we show that much of the divergence between different methods at different scales vanishes when we apply our improved dataset and statistical approaches. As mentioned earlier, site-level synthesis studies have been validated against even larger-scale top-down studies, so improved alignment between the national results of our component-level method and previous site-level synthesis results suggests much better agreement with top-down results[13,34]. Third, to isolate specific sources of disagreement between the GHGI and other studies, we reconstruct the GHGI emission factors beginning with the underlying datasets and uncover some possible sources of disagreement between inventory methods and top-down studies. Based on these results, we suggest a strategy for improving the accuracy of the GHGI, and likewise any country using a similar approach in reporting O&NG $CH_4$ emissions to the UNFCCC. We acknowledge that the results of our study required extrapolating relatively small sample sizes to the level of the US. Certain sources, especially tanks, are currently poorly characterized, and this prevents us from generating region-specific emission factor estimates. However, when evaluating our results, we must be clear that the baseline we are comparing to is not a world with perfect information about $CH_4$ emissions. It is the current GHGI, which is even more data limited.

## Results

**A new bottom-up approach.** Bottom-up approaches extrapolate component or equipment emissions rates to large (e.g., national) scales by multiplying emission factors (emissions per component or equipment per unit time) by activity factors (counts of components per equipment, and equipment per well) (Fig. 1). Our estimation tool requires two sequential extrapolations, first from the component to the equipment-level, and second from the equipment to the national or regional-level.

The approach utilized in our bottom-up estimation tool begins with a database of component-level direct emissions measurements (e.g., component-level emission factors). We generate component-level emission factor distributions for this study from

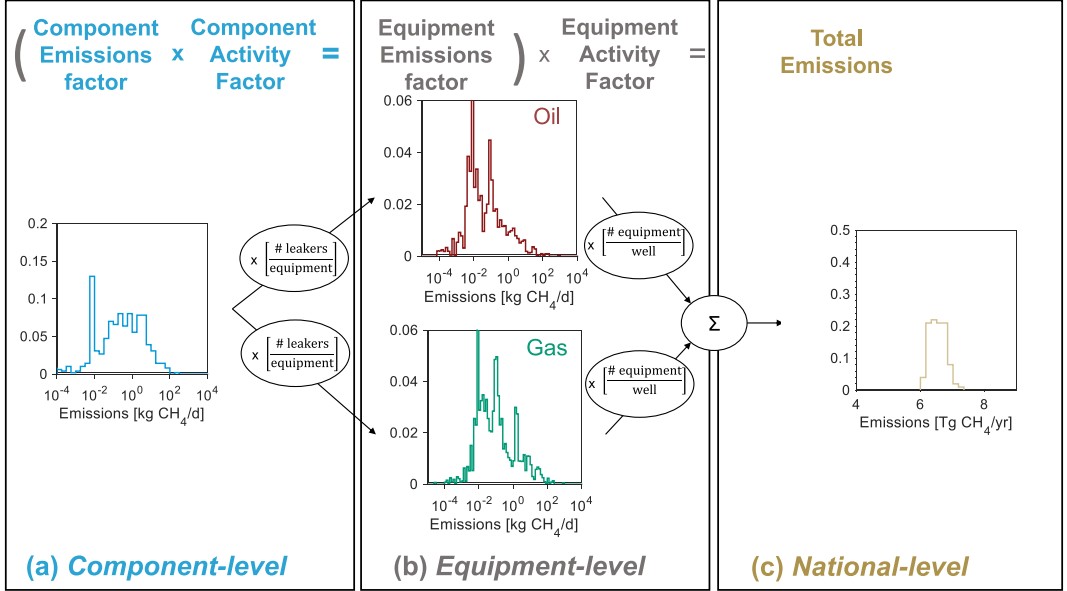

**Fig. 1 Schematic of this study's bottom-up CH₄ emissions estimation tool.** Calculation of total CH₄ emissions involves multiplication of emission factors (e.g., emissions per valve) by activity factors (e.g., number of valves per wellhead). Two sequential extrapolations are performed using an iterative bootstrapping approach. First, our database of component-level (e.g., valve, connector) emissions measurements (**a**) is extrapolated using component-level activity factors to generate equipment-level (e.g., wellhead, separator) emission factors (**b**). Second, these equipment-level emission factor distributions are extrapolated using equipment-level activity factors to generate a 2015 United States oil and natural gas production-segment CH₄ emissions estimate. This extrapolation is performed 100 times to generate a distribution of national-level CH₄ emissions (**c**) and estimate a 95% confidence interval (CI).

---

**Table 1 Summary of component-level datasets meeting inclusion criteria.**

| Study ID | Location | Number of quantified leaks | Number of components screened | Leak volumes used | Component counts used | Components screened |
|---|---|---|---|---|---|---|
| Allen[33] | Various | 646 | NR | Y | N | Various components |
| Allen[46] | Various | 378 | 378 | Y | N | Pneum. controllers |
| Bell[68] | Fayetteville | 247 | NR | Y | N | Various components |
| ERG[38] | Barnett | 1949 | NR[1] | Y | N | Various components |
| Thoma[69] | Uintah | 81 | 81 | Y | N | Pneum. controllers |
| Pasci[36] | Various | 192 | 54,618 | Y | Y | Various components |
| API[35] | Various | 251[2] | 102,680 | Y | Y | Various components |
| Clearstone[37] | Canada | | | N[3] | Y | |

Oil and gas methane emission measurement studies that reported raw data for quantified emissions measurements, fraction of components emitting, and component counts are summarized here. These studies are a subset of all studies that were examined closely, meeting inclusion criteria described. Detailed summary of each study's results are reported in Supplementary Methods 7.
NR not reported
[1]Screening counts are reported for several categories (connectors, valves, tanks) but counts are not comprehensive (see Supplementary Methods 4).
[2]Although only 251 data points from API 4598 were useful for quantification, 1780 leaking components were screened (i.e., only a subset of leaking components were quantified using the "bagging" technique).
[3]Given that leakage data were taken in Canada, we limit usage of this data to component counts.

a literature review building on prior work[11,30] and adding new publicly available quantified measurements (Table 1 in Methods). Our resulting tool's database includes ~3700 measurements from 6 studies across a 12-fold component classification scheme (see Supplementary Methods 4 for further description of this classification scheme). We applied emission factors as reported in the individual studies, with no modifications beyond unit conversion (noting that there are some differences between studies in High Flow Sampler bias correction for gas concentration and flow rate, which may introduce uncertainty to our results). Data for component counts and fraction of components emitting (the ratio of emitting components to all components counted) was scarce, with only 3 studies containing useful information for both ([35–37] for component counts and [35,36,38] for fraction of components emitting).

We derive equipment-level emission factors for our tool by random re-sampling (i.e., bootstrapping, with replacement) from our component-level database according to component counts per equipment and fraction of components emitting. Note that some of the cited studies will also calculate equipment-level emission factors. However, our study does not take the equipment-level emission factors as inputs. Rather, we take the combined component-level emission data, component counts, and fraction of components found to be leaking, therefore values calculated here will be different from the values calculated in those studies. Source-specific approaches were required for infrequent events (i.e., completions, workovers, liquids unloadings), methane slip from reciprocating engines, liquid storage tanks, and uncombusted methane from flare stacks (see Supplementary Methods 4 and 5).

We then perform a second extrapolation, using our equipment-level emission and activity factors to calculate a 2015 US O&NG production-segment $CH_4$ emissions estimate. For this step, our tool is integrated into the Oil Production and Greenhouse Gas Emissions Estimator (further description of OPGEE can be found in Supplementary Methods 4) and parameterized using 2015 domestic well count and O&NG production data (same dataset as Alvarez et al.[13]). A total of ~1 million wells and associated equipment are partitioned and analyzed across 74 analysis bins (Supplementary Methods 5). We performed a Monte Carlo uncertainty analysis repeating the bootstrapping algorithm 100 times across all ~1 million wells.

As both top-down and site-level measurement studies have demonstrated, there is a wide variability in $CH_4$ emissions across O&NG production regions[13,34]. Some of this variability will be captured through data sources and mechanics of our model, and some will not. As Omara et al.[34] demonstrate, a significant share of this variability can be explained by the combination of number of sites and natural gas production characteristics. Our model is able to replicate Omara et al.'s relationship between site-level productivity (Mscf site$^{-1}$ day$^{-1}$) and production normalized $CH_4$ (i.e., basins with low productivity sites demonstrate higher production normalized $CH_4$,[34] see Supplementary Fig. 12). We are also able to demonstrate a second trend from the site-level literature (e.g.,[39,40]) where emissions per site are higher at liquids-rich sites versus gas-rich sites (Supplementary Fig. 13, noting however that this trend is weak, and should only be considered suggestive). While we believe, based on these validation exercises that our model can describe variability across basins relatively well, we acknowledge that our results are still constrained by the

limited number of component-level measurement studies available. Beyond the production-related factors described above, variability will also be introduced by regulatory frameworks and operator practices that differ between regions. If data were available as a representative sampling of component-level measurements across basins, our method could capture this variability. However, given the data limitations, our measurements are biased towards certain geographies (e.g., tank measurements are sourced entirely from the ERG 2011 Fort Worth campaign[38]). As measurement campaigns progress over time, this issue should diminish.

**Comparison of US production-segment $CH_4$ emissions with site-level studies and the GHGI.** We first compare our resulting US 2015 O&NG production-segment $CH_4$ emissions estimate with the GHGI's estimate for 2015 produced in the 2020 inventory[25]. We also validate our bottom-up tool by comparing total emissions and emissions distributions with those generated in site-level synthesis studies. The total $CH_4$ emissions estimate of our model is compared with Alvarez et al.[13], and site-level distributions are compared with Omara et al.[34] (see description of site-level studies in Supplementary Methods 2 and methodological elements of the validation exercise in Supplementary Methods 5).

We estimate mean O&NG production-segment $CH_4$ emissions of 6.6 Tg yr$^{-1}$ (6.1–7.1 Tg yr$^{-1}$, at 95% confidence-interval, CI) (Fig. 2a, note that the CI only captures uncertainty due to resampling). Our mean, production-normalized emissions rate from the production segment is 1.3% (1.2–1.4% at 95% CI, based

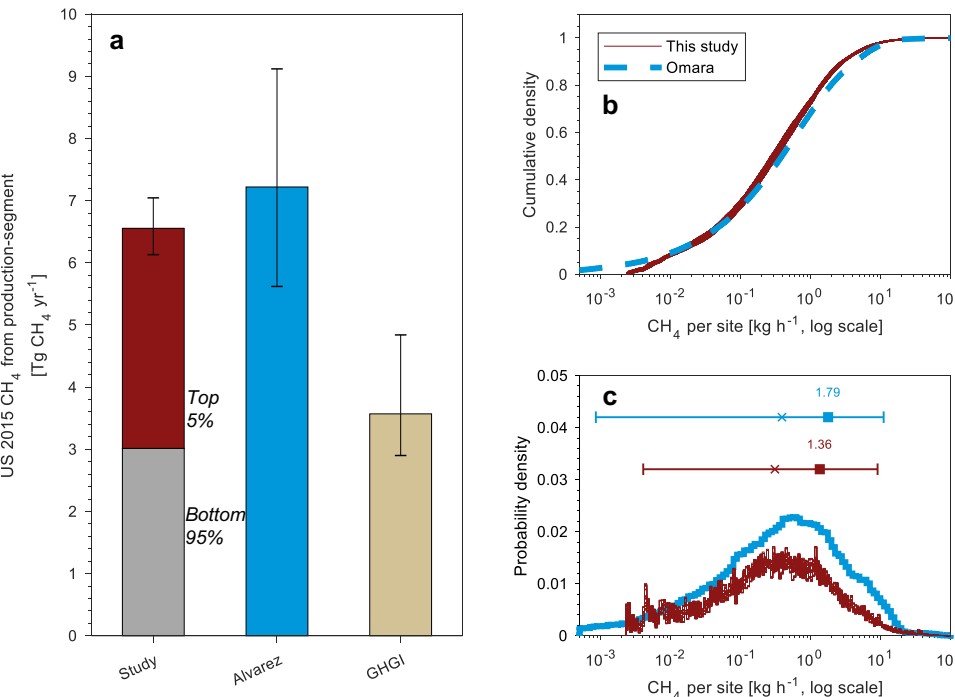

**Fig. 2 Comparison of results with previous site-level studies. a** Comparison of this study's aggregate estimate of United States 2015 $CH_4$ emissions from the oil and natural gas production-segment (mean of Monte Carlo uncertainty realizations) with site-level results of Alvarez et al. (see Table S3 in[13] minus contributions from offshore platforms and abandoned wells) and the Greenhouse Gas Inventory[25] including fraction estimated from super-emitters (top 5% of sources). Error bars reflect the 95% confidence interval based on the 2.5 and 97.5 percentile values extracted from the empirical distributions. We also compare probability distributions of our component-level simulations (red lines), aggregated into site-level emissions, with site-level results of Omara (blue line): **b** Cumulative distribution plot (CDF) describing the fraction of well-sites with emissions below a given amount, and (**c**) probability distribution of emissions rate per well-site with the mean (filled square), median (x), and 95% confidence intervals shown above the plots. Results of this study are presented using 100 Monte Carlo simulations. Because of the large number of sampled sites, the Monte Carlo simulations all converge toward the same size distribution in panels (**b**) and (**c**).

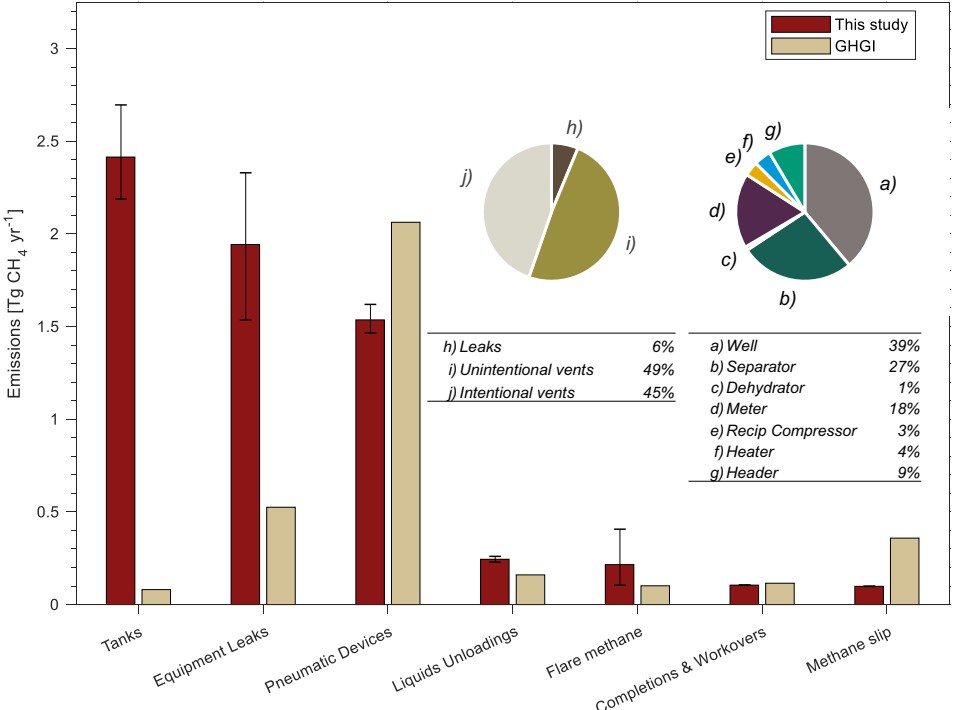

**Fig. 3 Source-specific CH₄ emissions comparison between this study and the 2020 Greenhouse Gas Inventory.** Bar chart compares $CH_4$ emissions estimates (mean of Monte Carlo uncertainty realizations) across source categories for the United States 2015 oil and natural gas production-segment between this study and the 2020 Greenhouse Gas Inventory (GHGI)[25]. Error bars reflect the 95% confidence interval based on the 2.5 and 97.5 percentile values extracted from the empirical distributions. Inset pie charts illustrate individual contributions of our inventory to equipment leaks (right pie chart) and tanks (left pie chart). Discrepancies with the GHGI are dominated by liquid hydrocarbon tank leaks, unintentional emissions from thief hatches and pressure-relief valves (PRVs), and flashing emissions (~2.3 Tg yr⁻¹ CH₄) and equipment leaks (~1.4 Tg yr⁻¹ CH₄). Details regarding the modelling of tank emission sources is given in Supplementary Methods 4. Results in tabular form are given in Supplementary Table 3 and Supplementary Table 4.

on gross NG production of 32 trillion cubic feet and an average $CH_4$ content of 82%[41,42]), slightly lower than Alvarez et al.[13], who estimate 1.4% (applying the same denominator as above). Both our bottom-up component-level inventory results and the Alvarez site-level results are approximately 2x those of the GHGI estimate of 3.6 Tg yr⁻¹ (year 2015 data[25], excludes offshore systems) for the O&NG production segment. Interestingly, the difference in US production-segment emissions between this study and the GHGI is approximately the same volume as our estimate of contribution from super-emitters (top 5% of emissions events). Given that our results match the Alvarez et al. site-level results, we conclude that the divergence between the GHGI and top-down/site-level studies is not likely to be due to any inherent issue with the bottom-up approach.

Figure 2b, c show that site-level distributions developed using our model match empirical distributions from the site-level synthesis study of Omara et al.[34]. To report our results on a basis consistent with site-level studies (recalling that sites can contain more than one well), we cluster equipment-level emissions outputs into production sites (Supplementary Methods 5). The tail of our modeled distribution closely matches the tail of the empirical Omara et al. distribution (Fig. 2b and Supplementary Fig. 35). This is of particular interest, given that recent papers assert the divergence between the GHGI and site-level studies is mostly due to an inability of the bottom-up methods to capture super-emitters[32,40]. Our results show that updated emission factors, through both more comprehensive datasets and revised modelling approaches, can recreate observed super-emitters.

Because our approach uses a component-level, bottom-up approach, we can investigate the source of differences with the GHGI. This cannot be done with site-level data. Relative to the

GHGI, contributions from equipment leaks in our estimate are larger by ~1.4 Tg CH₄ and tank leaks and venting by ~2.3 Tg CH₄ (Fig. 3). Together, these two sources contribute over half of total O&NG production-segment CH₄ emissions. The increase in estimated emissions from equipment leaks compared to the GHGI are due to our updated equipment-level emission factors; we know that the difference is not due to equipment-level activity factors because ours are nearly identical to the GHGI (see Supplementary Methods 3). Equipment-level emission factors are themselves a function of both component-level emission data and component counts, and we acknowledge that our model relies heavily upon the same early 1990s data set as the GHGI for component counts.

In the next section we will perform a deeper investigation into both component-level emissions data for equipment leaks and tank modelling as underlying contributors to differences between our results and the GHGI.

**Main sources of GHGI underestimation.** Given that our new component-level method is validated by the empirical results from site-level field studies, can we explain why the GHGI produces lower O&NG production-segment CH₄ emissions estimates? Results from our modelling (Fig. 3), in addition to recent revisions by the GHGI and other analyses ([33,43–46], see further discussion in Supplementary Methods 6), suggest that the downward bias of the GHGI is not primarily due to pneumatic devices, liquids unloadings, completions and workovers, methane slip from reciprocating engines, or uncombusted methane from flares (either the divergence is small, absolute emissions are small, or emissions are higher in the GHGI compared to our study). For

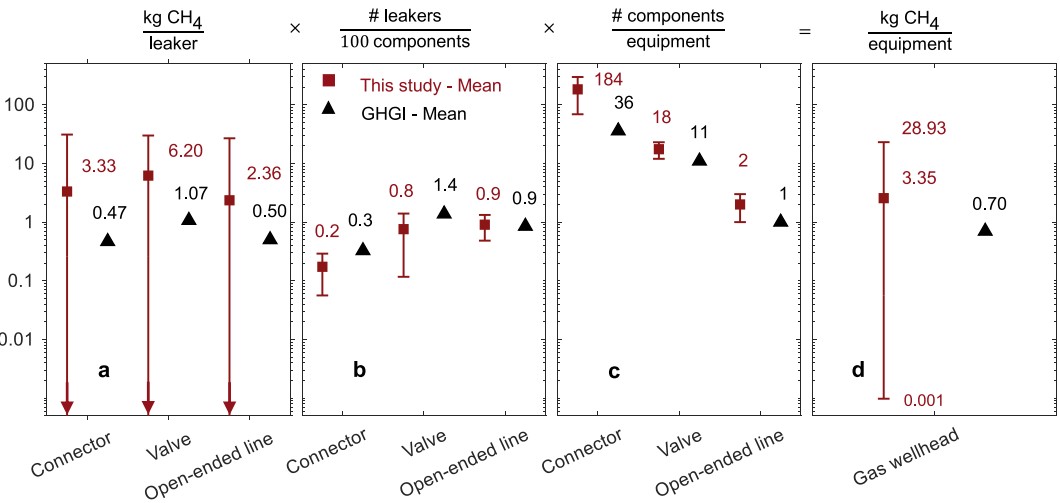

**Fig. 4 Example decomposition of the equipment-level emission factor for gas wellheads.** This study's equipment-level emission factor (**d**) for Western natural gas system wellheads is decomposed into constituent parts and compared with the Greenhouse Gas Inventory (GHGI). Error bars reflect the 95% confidence interval based on the 2.5 and 97.5 percentile values extracted from the empirical distributions and filled squares and triangles represent the mean. Constituent parts include component-level emission factors (**a**), fraction of components emitting (**b**), and component counts (**c**). When multiplied together, these factors have counteracting biases, with component-level emission factors and component counts contributing to higher emissions in our study versus the GHGI, and fraction of components emitting contributing to lower emissions in our study (Note that units differ for each panel, and also the logarithmic scale meaning that visible differences between points often span orders of magnitude). For illustrative purposes, there are several limitations to what is included in our decomposition plots. First, here we only show constituent data for Western natural gas systems; results for Eastern natural gas system are reported in Supplementary Methods 6 (Note that in actual usage in the GHGI, equipment-level emission factors for natural gas systems are a weighted average of both Western systems (API 4589[35]) and Eastern systems (Star Environmental,[47])). Second, we also limit this figure to connectors, valves, and open-ended lines (which account for the majority of components although our inventory and the GHGI also account for pressure relief valves, compressor seals, and other components in smaller numbers). Finally, decomposition plots are limited to component-level emission factors and fraction of components emitting at > 10,000 ppmv (this study) and pegged source factors (EPA GHGI) (see further discussion in Supplementary Methods 6).

these reasons, this paper focuses its analysis of the two largest sources of GHGI underestimation compared to our validated method: equipment leakage and liquid hydrocarbon storage tanks, whose emissions are 1.4 and 2.3 Tg CH4 lower than our estimates, respectively. See Supplementary Methods 1 for definitions of each emissions source.

The GHGI constructs emission factors for equipment-level leaks using an approach very similar to ours, where emission factors of individual components are aggregated according to estimated counts of components per piece of equipment. To explore differences in equipment leak estimates, we decompose equipment-level emission factors into the constituent parts: Component-level emissions data, component counts, and fraction of components emitting (the relationship between these parameters is defined in Fig. 4).

The GHGI further segments emission factors beyond petroleum and natural gas systems. Consistent with the underlying studies from the 1990s[35,47], GHGI equipment-level, equipment leakage emission factors for natural gas systems are subdivided by region (Western gas versus Eastern gas), and for petroleum systems data are subdivided by product stream (light oil versus heavy oil). Equipment-level emission factors for natural gas systems, for example, are a weighted average of both Western emission factors and Eastern emission factors. The GHGI approach to aggregating these factors to overall values for natural gas and petroleum systems is described in Supplementary Methods 6.

We demonstrate differences in equipment-level emission factors for equipment leaks via a decomposition into constituent factors for a single example (equipment type and region)— leakage from natural gas wellheads in the West (Fig. 4)—with equipment leaks from all other sources similarly described in the Supplementary Information (Supplementary Fig. 23 –31). The

difference between our study's equipment-level equipment leakage emission factor for Western natural gas wellheads and the GHGI—the difference to be explained by decomposition—is ~5× (3.4 kg day⁻¹ versus 0.7 kg day⁻¹). The underlying factors are plotted in Fig. 4.

First, we compare component-level emission factors, defined as the average emissions rate of leaking components (Fig. 4a). (Note that the average emission rate of leaking components is not the same as an average emission rate for all components). For Western natural gas and petroleum systems in the GHGI, component-level leakage emission factors are constructed using a method referred to by the EPA[48] as the EPA correlation approach (defined in detail in Supplementary Methods 6). In this approach, emission factors are constructed from a dataset of various facilities including oil and gas production sites, refineries, and marketing terminals ($n = 445$, data compiled in the EPA Protocol document[48]). The difference between our study's component-level emission factors and the GHGI for connectors, valves, and open-ended lines (the components comprising the wells) is ~7×, 6×, and 5× respectively (Fig. 4a). We can only speculate as to why this difference exists, but possibilities include sampling bias in the original collection process or fundamental differences in the populations sampled in the EPA's basis datasets versus those in this study (for example, most O&NG is now produced from unconventional shale formations whereas it wasn't during the time of the original GRI study). Note that the decomposition in Fig. 4a is limited to connectors, valves, and open-ended lines (which account for the majority of components) although our inventory and the GHGI also accounts for pressure relief valves, regulators, compressor seals, and other miscellaneous components in smaller numbers).

Figure 4b compares the fraction of components emitting (the ratio of emitting components to all components counted), while

Fig. 4c shows component counts (number of components counted per piece of equipment). These have offsetting effects, where component-level emission factors and component counts contribute to higher emissions in our study versus the GHGI, and fraction of components emitting contributing to lower emissions in our study. The resulting total emissions per well (Fig. 4d) are the product of these factors, summed across all components.

Similar results are found across all equipment categories compared to the GHGI. In general, in our dataset, component-level emission factors are higher (5× to 46× comparing our emission factors for connectors, valves, and open-ended lines across all GHGI categories, see Supplementary Fig. 22–30), the fraction of components emitting is lower (1× to 0.06×), and the number of components per piece of equipment is generally, but not always, higher (0.5x to 20x comparing our emission factors for wells, separators, and meters across all GHGI categories). Considering the decomposition presented here, along with the rest in the Supplementary Information (plus some discussion of smaller factors not described here), we can explain much of the overall underestimation of the GHGI compared to our results for the equipment leaks source category.

One source of the difference not illustrated in Fig. 4 between our study and the GHGI is related to how equipment-level emission factors in the GHGI (for NG systems) are a region-weighted combination of Western US and Eastern US factors. Component-level emission factors in the Eastern data (e.g., Supplementary Fig. 20) are significantly smaller compared to both this study and the EPA Western US data and are derived from an even smaller sample from the 1990s (~100 quantified leaks). Since these measurements were made, NG production in the Eastern US has grown from <5% to ~28% of total US production (Supplementary Fig. 15). It is finally worth noting that quantified emissions measurements (based on bagged measurements, and not those based on correlation equations) were included in this study's dataset. Although these measurements are small fraction (~7%) of our total dataset, the contribution is higher for specific components (Supplementary Fig. 14) emphasizing the importance of future data collection.

Equipment-level emission factors and total emissions for each equipment class are also presented in Supplementary Tables 3 and 4. Taken together, the gap between this study and the GHGI for equipment leaks is higher for natural gas systems (1.0 Tg) versus petroleum systems (0.4 Tg).

The second source of significant divergence between this study and the GHGI for US $CH_4$ emissions in the O&NG production-segment is with emissions from liquid hydrocarbon storage tanks. The EPA GHGI constructs storage tank emissions estimates using Greenhouse Gas Reporting Program (GHGRP) data. The GHGRP is a program which collects emissions data from industrial facilities, where requirements for natural gas and petroleum systems are specified by the Code of Federal Regulations Section 40 Subpart W[49]. Based on GHGRP data for storage tanks (see further description in Supplementary Methods 6), we decompose total emissions for the GHGI into tank counts and emission factors allowing us to draw comparisons to results from this study.

Before presenting our decompositions, it is worth noting two key differences in modelling of emissions from liquid hydrocarbon storage tanks between our study and the GHGI (see further description of how our model estimates tank emissions in Supplementary Methods 4). First, whereas our model is based on direct measurements, the GHGI is based on operator reported simulations from software programs such as API E&P Tank or AspenTech HYSYS[50,51] (or rather, simulated emissions which are a function of measured process parameters such as temperature and pressure, see 98.233(j) of[49]). Second, because

of these differing approaches, whereas our emissions are classified based on measurement source (e.g., vent stack, thief hatch, etc.) GHGI emissions are classified according to the simulated process (e.g., flash emissions). Because of these differences in emissions classification, comparisons between decompositions of our study versus the GHGI will be imperfect.

With this in mind, we define emission factors in our decomposition as the summation of intentional emission factors and unintentional emission factors (Fig. 5). Here, intentional (flash related) emission factors are based on direct emission measurements at the vent stack for our study, and simulations of uncontrolled and controlled tanks in the GHGI. Our comparison of unintentional emission factors is less precise. In the GHGI, unintentional emissions are limited to what is reported under the category of malfunctioning separator dump valves (although it is unclear if additional unintentional emissions are reported alongside flash emissions in the other tank categories, see Supplementary Methods 6). Conversely, unintentional emission factors in our study are based on direct measurements of emissions from open thief hatches, rust-related holes, and malfunctioning pressure-relief valves.

We demonstrate the decomposition in Fig. 5 for petroleum systems (see Supplementary Fig. 33 in the SI for natural gas systems). Note that flash emissions will only occur at uncontrolled tanks, while unintentional emissions from thief hatches, holes, or pressure-relief valves could occur at either controlled or uncontrolled tanks. Figure 5 (and Supplementary Fig. 33 in the SI for natural gas systems) demonstrate that, while several factors contribute to differences, difference in emission factors for various unintentional emissions sources (between both natural gas and petroleum systems) are the greatest source of difference between this study and the GHGI. Unintentional emission factors are the product of (i) average emissions rate per event, and (ii) frequency of unintentional emissions events per tank. Both of these values are approximately an order of magnitude higher for our study as compared to the GHGI, contributing to the nearly two orders of magnitude difference in total emissions.

Our findings suggest that both the magnitude and frequency of unintentional emissions sources could contribute to significant underestimation in the GHGI. Due to the limited quantified, component-level data available on tank emissions (based upon safety and accessibility issues) our tank emissions measurements come from a single study in a single geographic area (Eastern Research Group in the Barnett shale,[52]). Therefore, more studies are required to provide a comprehensive view of tank emissions. Although the ERG study benefited from unique site access granted by municipal authorities, future studies should prioritize access to tank walkways and consider pursuing additional measures to sample thief hatches, pressure-relief valves, and vent stacks (ERG document the use of extensions to the High Flow Sampler tubing to access out-of-reach components and large nylon bags to sample oversized openings such as thief hatches[38,53]).

However, while quantified emissions data for tank sources are scarce, the existence of unintentional emissions from tanks (due to open thief hatches, rust-related holes, pressure-relief valves, etc.) has been corroborated by numerous ground and aerial surveys[40,54–56]. Several of these studies are summarized in Supplementary Table 37. Taken together, these studies provide further evidence that: (i) high emissions events are frequently observed at storage tanks, not just from vents but also at open thief hatches, (ii) these high emissions events are common at both controlled tanks and uncontrolled tanks, (iii) the frequency (events/tank) of unintentional emissions events is much higher than the rate suggested by the EPA (2%, see Fig. 5c) for malfunctioning separator dump valves.

Equipment-level emission factors and total emissions for intentional flash emissions and unintentional emissions are also

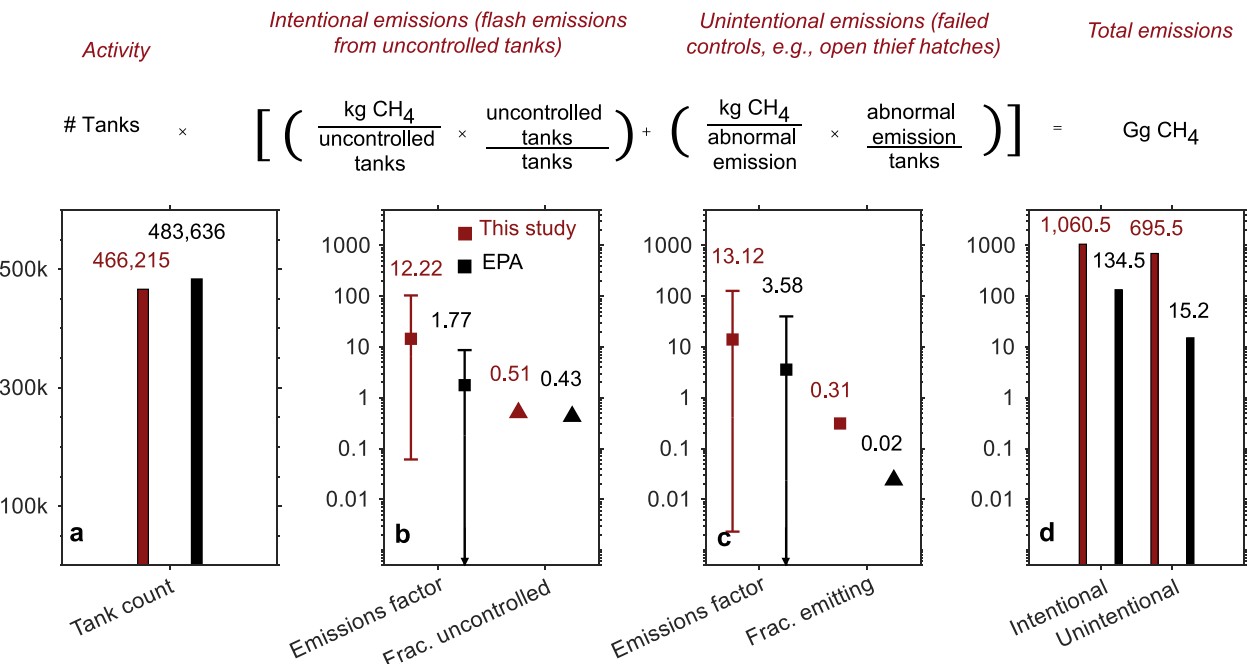

**Fig. 5 Example decomposition of total CH$_4$ emissions for crude oil storage tanks.** Total CH$_4$ emissions (**d**) for crude oil storage tanks in petroleum systems (for a decomposition of CH$_4$ emissions from condensate storage tanks in natural gas systems see Supplementary Fig. 33) are decomposed into several constituent parts and compared with corresponding factors in the Greenhouse Gas Inventory. Error bars reflect the 95% confidence interval based on the 2.5 and 97.5 percentile values extracted from the empirical distributions and filled squares and triangles represent the mean. Constituent parts include tank counts (**a**), the intentional emission factor (**b**), and the unintentional emission factor (**c**) (note the log scale for the right three panels). Intentional and unintentional emission factors are decomposed into emission factors (kg CH$_4$ per emitting tank) and control rates (fraction of total tanks emitting). Intentional emissions are defined as flash CH$_4$ released from uncontrolled storage tanks operating as designed. Unintentional emissions and the corresponding fraction-emitting value relate to emissions identified (at a screening value > 500 ppmv) at thief hatches, pressure-relief valves, and rusted holes. Note that, although both our activity data and the Greenhouse Gas Inventory activity data are based upon data from the Greenhouse Gas Reporting Program, our estimate of total tanks is different. This is because estimates of total well counts, which are used to extrapolate a population estimate for tanks, are slightly different (Supplementary Methods 5).

presented in Supplementary Tables 3 and 4. The gap between this study and the GHGI is much higher for petroleum systems (1.8 Tg) versus natural gas systems (0.5 Tg).

## Discussion

Development of accurate inventories at the equipment-level is critical for targeting CH$_4$ mitigation strategies. US government agencies[26], environmental groups[57,58], and researchers[59] rely on inventory data for policy design, cost analysis, formulation of leak detection and repair programs, and life-cycle assessment research. However, recent studies have emphasized a ~1.5×–2× divergence between the EPA GHGI estimates of CH$_4$ emissions from O&NG and those estimated from field measurements at different spatial scales. This suggests an opportunity for improvement in the GHGI approach.

In this study we develop a component-level, bottom-up approach validated by previous site-level estimates of US 2015 CH$_4$ emissions from the production segment of the O&NG sector. Consistent with site-level findings, our estimate is ~1.8 times that of the GHGI. The strength of our approach is that by developing our estimate using component-level data, we can diagnose at the equipment-level the key sources contributing to the GHGI underestimation. Our detailed decomposition identifies (i) underlying equipment-leak measurements and (ii) neglect of the contribution of unintentional emissions events at tanks (e.g., liquid hydrocarbon storage tank thief hatches) as likely the most important contributors to the underestimation.

By collecting and synthesizing all available component-level measurement data into a singular database, we believe this study provides a clear assessment of CH$_4$ emissions from the US O&NG production segment. Pooling of studies was necessary, given that research on super-emitters has demonstrated that "larger sample sizes are required … to achieve targeted confidence intervals"[30]. However, as we have described, our data may not adequately represent all regions of the US, especially for certain source categories. Sub-sampling in our larger dataset to focus on particular regions or types of facilities may offer spurious improvement, wherein specificity for that region or type of facility may be improved, but generalizability is hindered because the sample sizes for each new sub-sample become small. Future research should target data collection to fill these gaps in the literature to improve size and representativeness of samples. In addition, we note that this study's approach of incorporating data across multiple studies could challenge a preference of inventory administrators to evaluate the accuracy and representativeness of original data sources on a study-by-study basis.

These results demonstrate that the bottom-up methodology is a valid approach to produce accurate emissions estimates and that improvements to inventory methods are possible through both more comprehensive datasets and revised modelling approaches (demonstrated through respective contributions to the decompositions in Figs. 4 and 5). For development of emission factors for equipment leaks, this study applies a very similar approach to the GHGI, but with a new dataset of component-level emission factors, fraction of components emitting, and component counts.

Thus, differences can be largely attributed to data sources. Since our dataset is larger and contains more recent measurements, we suggest that it is likely to be more representative of today's conditions. For development of emission factors for crude and condensate storage tanks, differences are believed to be largely a result of the GHGI neglecting emissions from failed tank controls (e.g., open thief hatches). Although we attempt to estimate their contribution, and reference supporting site-level surveys, tank emissions remain a significant data gap. Given that locations of emissions sources from tanks are fewer (i.e., only possibilities are vents, PRVs, and thief hatches) compared to other equipment, site-level measurement campaigns (e.g., helicopter or airplane) could serve as more straight-forward alternatives to onsite measurement (which are particularly challenged for tanks that pose safety hazards and require access privileges). Such campaigns should be designed to refine the accuracy of the fraction and magnitude of unintentional emissions.

Because all emissions data and activity factors (with some exceptions, noted in methods) are US-based, emission factors from this study (summarized in Supplementary Table 2, 3 and 4) could be implemented in US inventories. Emission factors for equipment leaks could be implemented relatively easily by updating existing sources categories. Implementing emission factors from storage tanks based on this study would require modifications to source categorization, for example, through the addition of a new factor to take into account failed controls like open thief hatches. Regular efforts to validate equipment-level emission factors by comparing existing or new emission factors with measurements from randomly sampled sources at different spatial scales (i.e., validating component-level, direct measurement campaigns with downwind truck or airplane-based measurements) would also improve accuracy and build into inventory efforts the ability to correct data over time.

The results of this study are also relevant globally, both as inputs to default emission factor databases and as a generalized methodology for generating emission factors in different countries. All parties to the UNFCCC submit annual inventories, generated using a bottom-up approach, to report on progress towards GHG targets. The IPCC's Guidelines for National Greenhouse Gas Inventories outlines three approaches towards producing an inventory, with the simplest approach (Tier 1) based on IPCC default emission factors[27,60]. Default emission factors for the petroleum and natural gas systems production-segment are in some cases based upon the same underlying data sets as the GHGI[60]. This means that, in addition to the US-submitted GHGI, other countries using Tier 1 emission factors will be contributing $CH_4$ estimates according to data that we have found likely to be underestimating of actual emissions. Recommendations offered herein, if implemented, may improve emissions estimates globally. Given the sparsity of data globally, we are unable to state how much error is introduced by use of these factors globally.

It should be noted, however, that at the time of writing of this publication IPCC Tier 1 emission factors are unlikely to be updated soon. For agencies wishing to improve the accuracy of Tier 2 emission factors this study identifies sources towards which efforts should be focused (some countries, e.g., Canada and Australia[37,61], have requisite component-level data). We believe that incorporation of a larger emissions dataset and revised modelling approaches to sources including storage tanks and flaring has produced a more accurate inventory estimate for production segment $CH_4$. Finally, although our focus in this paper is on inventory development, the results of this study will also be relevant to industry in targeting and prioritizing practices to reduce $CH_4$ emissions.

## Methods

Here, we describe the methodological aspects of each of this study's three key contributions: (i) tool development, (ii) generating a US $CH_4$ estimate for the O&NG production-segment, and (iii) decomposing GHGI emission factors. Our methods are also described in greater detail in the Supplementary Information.

**Terminology**. To avoid confusion, we do not use the term fugitives. To the extent possible, this study adopts the terminology conventions of the GHGI and the GHGRP with equipment leaks and vents (see further discussion in Supplementary Methods 1).

**Tool structure**. The analysis platform for this study is the $CH_4$ emissions sub-routine embedded within the Oil and Gas Production Greenhouse Gas Emissions Estimator (OPGEE version 3.0). This subroutine processes equipment-level emissions distributions and well and production values and produces gross emissions estimates.

The following equation describes the $CH_4$ emissions subroutine:

$$Q_{population} = \sum_{i=1}^{n_{field}} \left\{ \sum_{j=1}^{n_{wells,i}} \left[ \sum_{k=1}^{n_{equip}} EF_{i,j,k} * af_k \right] \right\} \qquad (1)$$

Here, a field represents a subpopulation (or bin) of wells that share similar production characteristics (e.g., gas-to-oil ratio). This binning was necessary because OPGEE generates outputs (carbon intensity or $CH_4$ rate) on a field basis. For each field, $i$, emissions are calculated well-by-well. For a single well, $j$, equipment-level emissions are calculated by multiplying a randomly drawn emission factor, $EF_{i,j,k}$ (kg equipment$^{-1}$day$^{-1}$), by its respective activity scaling factor, $af_k$ (equipment well$^{-1}$). Because we iterate across wells, there is no need to explicitly multiply the activity scaling factor by well count (see Supplementary Methods 4). Emissions are calculated across all equipment classes, $k$.

**Database on component level studies**. Equipment-level emission factors are generated using a component-level measurement database. We conducted a detailed literature review to inform the database for this study. This review built on prior work done for Brandt et al.[11,30] and adds new publicly available component-level measurements. Studies were reviewed for information regarding: (i) data on quantified emissions volumes per emitting component or source, (ii) activity counts for numbers of components per piece of equipment or per site, and (iii) fraction of components found to be emitting in a survey.

Quantified emissions data were further filtered for: (i) data collected within the production (upstream) segment, (ii) and data collected in the United States (although we do include some component count and fraction leaking data from Canada, see further details in Supplementary Methods 4). A total of 6 studies and ~ 3,700 measurements met our inclusion criteria (see Table 1).

To aggregate the data from the various studies, we developed 12-category and 11-category classification schemes for components and equipment, respectively. For components these include: Threaded connections and flanges, valves, open-ended lines, pressure-relief valves, compressor seals, regulators, pneumatic controllers/ actuators, chemical injection pumps, tank vents, tank thief hatches, tank pressure-relief valves, and other (miscellaneous) components. For equipment these include: Wells, headers, heaters, separators, meters, tanks – leaks, tanks – vents, reciprocating compressors, dehydrators, chemical injection pumps, and pneumatic controller/actuators (note that the "tanks – leaks" category tracks all non-vent/hatch emissions on a tank, e.g., connectors, valves, etc., while the "tank – vent" category tracks all vent/hatch related emissions).

To align the categories of components used by the authors of a study to our common component definitions, we create a set of correspondence matrices to perform consistent matrix transformations (see Supplementary Methods 4).

In addition to component-level emissions measurements, we also require component counts and fraction of components emitting. A total of 3 studies contained information on component counts[35–37], and we aligned the data into our standard categories. Data on fraction of components emitting was also scarce, with 3 studies containing useful information[35,36,38]. The fraction emitting rate is an important parameter in deriving equipment-level emission factors but varies greatly by study due to (i) differences in screening methods between studies (e.g., Method 21 vs. infrared camera) and (ii) use of different screening sensitivity to assign a component to the emitting state (10 ppmv vs. 10,000 ppmv). Therefore, based on the technologies employed, different studies may be sampling different parts of the true population emissions distribution. To ensure that we are not over or under-sampling a subset of the true distribution, we split our dataset at 10,000 ppmv (see reasons for this threshold in Supplementary Methods 4). Different quantified emissions bins and fraction emitting values were derived for the two halves.

**Equipment-level emission factors**. We required a variety of approaches to describe the different sources of emissions. The most common approach taken by this study, utilized for equipment leaks and unintentional vents, is the stochastic failure approach. In the stochastic failure approach, we combine component-level emissions data, component counts, and fraction emitting values to produce equipment-level emission factors. These emission factors take the form of

distributions which are generated by iteratively resampling our emissions datasets (see Supplementary Methods 4).

For each equipment category, we iterate across component categories and draw emissions measurements according to a probability specified by the fraction emitting value. Given that we split our dataset at 10,000 ppmv (describing quantified emitters that were missed by optical gas imaging but detected with Method 21 below the threshold, and emitters that were caught with optical gas imaging above the threshold), we develop two sets of emission factors. These two emission factor distributions are superposed to form our best approximation of the true emissions distribution (Supplementary Methods 4).

We applied separate approaches for flashing emissions from tanks, methane slip from reciprocating compressors, and intermittent and startup losses from liquids unloading, completions, and workovers. These approaches are described in Supplementary Methods 4.

**Equipment-level activity factors**. In the GHGI, direct equipment counts are not available for every year. As an approximation, the GHGI uses activity drivers such as gas production, number of producing wells, or system throughput. Activity drivers are multiplied by a scaling factor (e.g., separators per well) derived from a subsample of the population. For each piece of equipment, we employ well counts as the activity driver. Since the 2018 GHGI, the EPA has calculated activity factors for most equipment using scaling factors based on GHGRP data. Scaling factors based upon reporting year 2015 equipment counts are multiplied by year-specific wellhead counts to calculate year-specific equipment counts[62].

**Development of representative fields for analysis**. In OPGEE, fields are described with over 50 primary input parameters, and numerous secondary parameters. Given that we are restricting our analysis to $CH_4$ emissions in the upstream sector, however, we only concern ourselves with a handful of inputs: Oil production, well count, gas-to-oil ratio (GOR), and methane mole fraction. The 2015 well count and production data (Supplementary Table 15) were based on the dataset from Alvarez et al.[13], which were originally derived from Enverus and filtered to remove offshore and inactive wells (~6,000 wells removed).

The total well count according to the Alvarez et al. Enverus dataset (1,005,191, see Supplementary Table 15) is ~15,000 wells lower than the estimate of the EPA[25]. We discuss possible reasons for this difference (Supplementary Methods 5), but overall a difference of ~1.5% in well counts will not significantly affect our $CH_4$ emissions results.

In order to account for the heterogeneous nature of O&NG systems, the total population was divided into several simulation sub-populations (or bins) according to the production GOR (where gas wells have a GOR > 100 mscf bbl$^{-1}$, [63]), gas productivity, and liquids unloading method. 60 bins were developed for natural gas systems while 14 bins were developed for petroleum systems (Supplementary Methods 5).

When OPGEE iterates across each bin of wells, a conservation of mass (COM) conditional statement is implemented to ensure that the summed emissions do not exceed gas production (also accounting for the gathering and boosting, processing, transmission, and distribution sectors, see description of algorithm in Supplementary Methods 4). Note that the COM check is required because, unlike the site-level data from Omara et al.[34], few component-level measurement studies provide well-level meta-data (e.g., well liquid and gas production, well age, etc.) with associated emission measurements. Therefore, although well characteristics are binned for OPGEE, each bin draws upon the same sample set of emission measurements. Thus, in some instances, OPGEE can draw a leak that is larger than the volume produced, violating COM. These draws are rejected and redrawn to ensure COM.

**Uncertainty analysis**. This study applies the Monte Carlo method to estimate uncertainty. Input parameters—component-level emission factors, component counts, and fraction of components emitting—are assigned distributions, and the range of uncertainty in these distributions is propagated through the model. Therefore, the full range of uncertainty is captured to the extent that these distributions encompass the full set of possible values.

A single OPGEE simulation will produce an estimate of total US $CH_4$, but it will not output a distribution. We run OPGEE 100 times (100 Monte Carlo iterations), each using a different set of equipment-level emission factor distributions (further description in Supplementary Methods 5). In producing variable equipment-level emission factor distributions, component counts and fraction of components emitting are approximated as uniform distributions between the maximum and minimum values found in our surveyed studies (see Supplementary Table 6 and 7 for component counts and Supplementary Table 11 for fraction leaking). Unfortunately, sparse available data do not allow us to determine a likely distribution shape for these parameters.

**Comparison with the EPA GHGI: Equipment leakage**. The construction of equipment-level emission factors in the GHGI is rooted in several studies conducted in the 1990s. We review these studies and trace how emission factors in today's GHGI are derived from these earlier analyses. The modelling approach of the early 1990s studies is closely related to the approach in this paper, in that

equipment-level emission factors are calculated from component-level emissions measurements and counts. By gathering the underlying datasets used to construct the GHGI's equipment-level emission factors we can generate component-level distributions for comparison with the distributions of our study.

The GHGI relies on a 1996 report by the Gas Research Institute ([64], henceforth referred to as the GRI report) for natural gas systems and a 1996 calculation workbook by the American Petroleum Institute ([65], henceforth referred to as API 4638) for petroleum systems. These reports were not measurement campaigns, rather these reports summarized the results of multiple earlier works. The GRI report references API 4589 ([35], sites 9–12) for the Western US natural gas system and Star Environmental[47] for the Eastern US natural gas system. API 4638 references data from API 4589 (sites 1–8). Therefore, only two measurement campaigns underlie GHGI equipment leakage: the API 4589 and the Star Environmental datasets.

We first analyze the screening data in API 4589 and Star Environmental and follow the methodologies outlined in Supplementary Methods 6. In API 4589, screening concentrations from Appendix C were scanned and tabulated. Unfortunately, it was not possible to re-derive the component-level emission factors in the Star Environmental dataset. This was for two reasons. First, in the Eastern leak quantification data (provided in Appendix F,[47]), information is not provided on components measured. Therefore, quantified emissions cannot be connected to the screening values contained in Appendix E. Second, the Eastern dataset does not report how they assigned leak volumes to the 81 instrument readings > 10,000 ppmv which were not quantified with the Hi Flow sampler. Therefore, component-by-component distributions can only be generated for API 4589.

After digitization and re-engineering of the GHGI methods, we can compare the distributions of the resulting component-level estimates with our dataset (Fig. 4, with additional comparisons in Supplementary Methods 6).

**Comparison with the EPA GHGI: Tank emissions**. To reconstruct emission factors for crude and condensate storage tanks, we begin by downloading GHGRP data from the "Envirofacts GHG Customized Search" tool[66]. After gathering the data, we segment the dataset according to product stream (natural gas, petroleum systems) and tank class. However, before making any comparisons with this study, we need to adjust how emission factors are reported by the GHGI. The GHGI reports storage tank emission factors on a throughput-basis (kgCH$_4$ bbl$^{-1}$ year$^{-1}$) and our study reports emission factors on a tank basis (kgCH$_4$ tank$^{-1}$ day$^{-1}$). Fortunately, in addition to tank throughput, atmospheric storage tank counts per sub-basin are also reported to the GHGRP by tank class.

Emission factor distributions (Fig. 5) are calculated by dividing total emissions by tank count for every sub-basin (or row in the downloaded dataset). In Supplementary Methods 6, we validate this approach by calculating and comparing throughput-basis emission factors with those reported in the GHGI.

## Data availability
The datasets generated and analyzed during the current study are available in a Github repository[67]. Certain datasets used are propriety and not publicly available. These include the Enverus dataset, used to generate well count and production parameters, and the Wood Mackenzie dataset, used to generator gas-to-oil ratios for oil-only wells.

## Code availability
The OPGEE 3.0 model and supporting code are available in the same Github repository[67]. Descriptions of the model are found at both the Github repository and the current study's supplementary information.

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

## Acknowledgements
This work was funded by the California Air Resources Board, grant 18ISD011. Support for the work was also provided by Novim under a Limited Sponsorship Agreement with the Joint Institute for Strategic Energy Analysis of NREL. This work was authored in part by the National Renewable Energy Laboratory, operated by Alliance for Sustainable Energy, LLC, for the U.S. Department of Energy (DOE) under Contract No. DE-AC36-08GO28308. Funding provided by Novim under a sponsorship agreement with the Joint Institute for Strategic Energy Analysis. The views expressed herein do not necessarily represent the views of the DOE, the U.S. Government, or sponsors. The authors would also like to thank Gregory Von Wald, Kyle Pietrzyk, and Dante Orta Alemán for assistance with model simulations.

## Author contributions
A.R.B, G.A.H., D.C., J.E., and J.S.R conceptualized the study. J.E. and A.R.B. developed the original model. J.S.R. improved upon the original model, implemented the model in the Oil Production and Greenhouse Gas Emissions Estimator, and applied the model to this study. Q.L. advised on model implementation. D.L. and M.O. contributed datasets. J.S.R., E.D.S., and A.R.B. drafted and finalized the manuscript. A.P.R., G.A.H., J.E., D.L., M.O., and Q.L. advised on analysis and revised the manuscript.

## Competing interests
The authors declare no competing interests.
