## [Peer Review File · Nature Communications]

REVIEWER COMMENTS

Reviewer #1 (Remarks to the Author):

The authors construct a bottom-up emissions estimate and attempt to reconcile differences between EPA's greenhouse gas inventory with recent top-down studies that find much larger methane emissions from the oil and gas sector. I think this manuscript is well-written, the analysis is sound, and the article makes an important contribution to the field. I recommend the manuscript for publication. I have a few relatively minor comments and suggestions:

Line 80-81: Is there a reference for this statement? In theory, many atmospheric observations would reflect the integrated signal of emissions that occurred over many hours or a few days (depending upon the measurement).

Line 141: Why apply the bootstrapping algorithm 100 times? If the distribution of emissions from wells has a heavy tail and if you are particularly interested in extreme values, then it may be important to run more than 100 simulations to full capture that distribution. It's not uncommon to run $1e3$ or $1e4$ Monte Carlo simulations, depending upon the complexity of the distributions involved. Do you think that 100 simulations is sufficient?

Line 222: Are there studies of facility-level or component-level measurements that reached parallel conclusions?

Fig 4: What do the vertical maroon bars represent? These bars are quite large and encompass the EPA estimate. Hence, I think it would be a useful point to explain to the reader.

Line 375: I think the quotation mark is in the wrong place.

Line 394: Can you clarify what you mean by different spatial scales? Are you referring to atmospheric measurements and satellite measurements (i.e., TROPOMI), facility-level measurements, component-level measurements, or something else?

Reviewer #2 (Remarks to the Author):

This is a very thorough and novel effort. The major claims of the paper are that the observed gap between top down studies and the U.S. GHG Inventory is not due to an inherent issue with bottom-up approaches, that the gap can be considered to be primarily due to underestimates in equipment leak and tank emissions, and that improved modeling of equipment leak and tank emissions can address the gap. These are novel claims and will be of great interest to the community. The paper will likely influence thinking and potentially, future research efforts. Because of the potential influence, it is important that additional clarification and information is provided to either strengthen the conclusions or enhance transparency about caveats and potential alternative explanations for the gap.

The authors have conducted a comprehensive review of available data and compiled a very useful database. Data in both the GHGI and the alternate data sets used in this study are somewhat limited (as only limited data

exist). Use of the alternate data sets addresses the gap, but if those data sets themselves are inaccurate or not representative of national emissions, then this paper could be incorrectly eliminating other potential sources of the gap, and influencing research efforts to move away from those and towards leaks and tanks. For example, for tanks, the key data set with year 2011 measurement data incorporated into the model is limited to one region. It is unknown how representative 2011 tank emissions in that region are of tank emissions in the U.S., and therefore how informative that data set is to understanding national-level gap. For equipment leaks, some of the key input data for the equipment leak model come from studies that calculated lower leak emissions per equipment than the GHGI (Allen et al. and Pasci et al.) and some of the input data come from the same studies used to develop the GHGI (STAR). The authors could provide additional information on how the deconstructed data from those studies resulted in higher emissions than the GHGI in their model. The other input data for equipment leaks come from a Canadian study, which the authors note may be impacted by different regulations/practices. Information on any QC performed on the model would also be helpful.

It is unclear from the information presented to what extent the gap is addressed through use of different input data versus through a different modeling approach. More information on how the study input data (e.g. a table with average EFs) compare with the GHGI will help strengthen the argument that it is the modeling approach that is addressing the gap versus different emission factor data.

The paper considers the gap to be closed based on developing similar emissions estimates using component level to those developed with site-level data. Additional information about the site level study (e.g. how many sites were measured by the site level campaigns, how much variation there is in those measurements, were the sites oil versus gas, high or low production, geographic scope) and how the comparison was done (e.g. be clearer that this study did not develop estimates for the specific sites measured in the site-level campaign) would strengthen the paper.

It would be helpful to provide clearer information on the relative contribution of petroleum production versus natural gas production for the gaps in tank and leak emissions. It appears that the tank difference is largest for petroleum systems (model is around 1,400 Gg larger than GHGI for oil, and around 600 Gg larger than GHGI for gas), and that for leaks its larger for gas (1,000 Gg larger than GHGI for gas, and 500 Gg larger for oil). Also, my sense is that emissions data from oil production sites is far more limited than from natural gas production sites. If this is the case, the paper should be clearer about this—are the conclusions on the sources of the gap stronger for natural gas than for petroleum?

The authors recommend applying a similar method to that used in this study to develop the U.S. GHGI and also GHGIs internationally. Additional rationale for this recommendation should be provided--given the uncertainties and the limited input data, what are the benefits of applying such a method to develop an annual GHGI versus instead applying EFs developed from such an analysis (or instead just applying average EFs from input studies)?

Finally, the authors have developed a unique understanding of the available data that can be used to develop GHG Inventories, and how limited that data can be. Many research efforts on methane do not produce data that can be used in the type of inventory analysis conducted by the authors or do not focus on emission sources most likely to contribute to the gap. If appropriate, recommendations on future research priority areas based on the authors' experience could enhance the influence of the paper.

Line by line comments:

Main text:

103: Explain the relative contribution of each (improved dataset and different statistical methods) for closing the gap.

104-105: The comparison between site-level results and the study's results could be explained more clearly. My understanding is that there was no modeling done in this study for the sites that were measured for comparison so it's more that a modeling of national level emissions using site-level measurements was compared with a modeling of national level emissions using component-level measurements.

123: Specify the timeframe over which these data were collected.

129: Suggest providing information here the representativeness of the component-level EF dataset. For instance, are the data from different regions and different time periods?

188: Is this a recommendation for how to compile the GHGI or a recommendation for how to develop EFs for GHGIs?

242: These distinctions are made in the GHGI calculations. EPA provides results on its website at a higher-level but can also provide the input information.

299: Is there a recommendation for how to incorporate these components into GHGI?

313: Are these tank emission modeling programs based on direct measurements though? If so, is this then comparing one way of incorporating direct measurements into a model with another? Are there any readily identifiable specific issues with the models used in GHGRP?

342: This limitation of the tank data is an important caveat that should be discussed in more detail. There is anecdotal and observational information and (limited) measurement data indicating that the GHGI underestimates tank emissions. However, closing the gap for tanks here seems to have been done by assuming that a limited number of measurements in a single region is correct (or at least more correct) and that the national level estimates based on facility-reported tank modeling is incorrect. Based on available data, the approach makes sense, but the authors should be clearer that there are uncertainties here.

344: If possible/appropriate, build out recommendations here. It seems that truly closing the gap would require a lot of tank measurements, which are challenging. If the ERG study approach is a good model, note that.

358: It's surprising that the study has a different fraction for uncontrolled tanks than the GHGI (which uses GHGRP data). There is not another dataset for fraction of control with as much coverage as GHGRP. Please clarify the source of this fraction.

357: In the Dallas ft worth study, storage tank breathing and standing losses were not measured/calculated. Non routine emissions also were not included. Does that mean that that study could be an underestimate itself? The study also used EPA correlation equation for tank calcs.

368: Explain the validation in more detail. Were the specific sites measured by Omara et al modeled here?

375: Consider expressing as "as likely the most important contributors"

398: This line seems to be referring to the 2006 IPCC Guidelines for National Greenhouse Gas Inventories. The title should be corrected. Note that there are several versions of the IPCC guidance. The 2006 guidance likely (documentation not always clear) includes U.S. and Canadian data as input to default factors.

404: The 2019 Refinement is a final document, and updates to the U.S. GHGI do not carry through to the IPCC guidance.

405: Instead of the study influencing IPCC Tier 1 EFs, which is not possible in the near future since the guidance was most recently updated in 2019, the authors could instead note its relevance for countries wishing to develop country-specific Tier 2 factors.

406: Change text to UNFCCC inventory reporting. The methods come from IPCC; the reporting requirements (and specification of which version of IPCC GL to use) comes from UNFCCC. The authors could consider also

noting that detailed emissions data may be helpful for countries with NDCs including oil and gas and this approach could contribute towards the development of robust source-specific estimates and track emissions over time.

408: Provide a stronger argument that it's the approach versus the input data that's providing the improvement. Especially if the recommendation is meant to apply to other countries—data are extremely limited in other countries. Is a better recommendation to focus measurement efforts on leaks and tanks?

418: To be clearer and improve international relevance, it would be helpful to adopt IPCC terms (fugitives = emissions from leaks, venting and flaring).

463: 51 is missing from the table

504: Enverus continually updates its well counts. It may be better to use latest Enverus counts for 2015. Another option is to add a sentence on how the total well counts in the Alvarez 2015 estimate compare with the 2015 well count estimate in the 2020 GHGI.

509: Provide additional explanation on how the authors were able to match EFs, component counts, and leak frequency to the different GOR levels, given how limited data are. Are data for certain GOR groups more limited than others?

510: Provide additional explanation on the 60 bins of data, given how limited data are.

697: This was probably meant to be a reference to the Energy volume of the 2019 Refinement (not the forestry volume).

Supplemental:

52: For transparency, it would be helpful to see a table with the average GHGI EF, the average EF from this study, and the average EF from all of the input studies. This will help give a better sense of whether it is the approach or the use of a different data source that is driving the difference.

174: If there aren't oil EFs in this study, how was the model run? How was the comparison made? Is most of the gap oil? If so, what evidence is there? For subpart W columns, provide footnotes explaining the difference between leaker methods i and ii. How are the GHGRP splits between east and west taken into account?

185: Were the 2015-specific emissions data used for tanks?

188: Provide additional information on the approach for assigning EFs/component counts to marginal wells. How many data points are included in each category? Do all of the input studies provide information on production level from the wells measured?

195: Why are the completions and workovers emissions so different from GHGI? Shouldn't these be the same or at least more similar since they are using the same data source? Does this indicate an issue with the modeling?

195: What is the combustion emission source? In GHGI, combustion emissions are included in a different section of GHGI (fossil fuel combustion). Is it methane slip?

217: Since the GHGI was updated to use GHGRP data, this is no longer relevant. The Marchese adjustment was from before the GHGI had data (from GHGRP) to allow for a distinct gathering segment and instead had any gathering emissions included in with onshore production.

220: The Allen et al. counts of pneumatics per well were from a far smaller sample size compared to GHGRP, which has pneumatic controller counts reported every year from hundreds of thousands of wells. Consider updating the model to use the GHGRP data set.

309: Here it is noted that few if any component-level studies break out data by production volume. How/why then is the model breaking the data into groups based on production?

346: Input data from Canada are included. Note the exception here (leak counts I believe) and how it effects the conclusions--here the authors say that differing regulatory and operating standards could impact results; yet the Canadian data are included for equipment leaks where one of the key discrepancies was observed-- what role did the use of Canadian data potentially play?

457: Clarify how oil versus gas and different production level EFs were assigned, considering the limited data availability.

536: Recommend using the GHGRP data on pneumatic controller counts

634: Does this approach potentially underestimate the EF for marginal wells? E.g. leak detection may be less prevalent at marginal wells so it may not necessarily be the case that they have much lower emissions than nonmarginal wells.

636: How are the means higher than the two subcategories?

670: NSPS does not require controls at all tanks. It requires controls at new or modified tanks over a certain threshold.

693: Note that the # of wells onsite is lower here than for many site-level studies. What does this mean for comparability of the results with other studies?

724: Provide rationale for why Allen et al was used for liquids unloading as opposed to GHGRP.

791: Why are the values for completions and workovers so different from GHGI if they are using the same dataset?

793: The emission factors do not seem to match those used in the GHGI.

810: See previous comment—Marchese adjustment no longer needed.

821: The equipment should all be allocated to production.

821: Recommend use of GHGRP for pneumatics per well for reasons noted above. In addition, this may be a misinterpretation of the Allen et al counts. A previous EPA assessment estimated 1.5 venting pneumatics per well based on Allen et al.: “The Allen et al. study observed 2.7 controllers per well surveyed. Only sites with venting pneumatic controllers were visited for the study. However, the study notes that some sites use non-venting and/or non-pneumatic controllers, and that therefore 2.7 controllers per well represents an upper bound. To develop rough estimates of national emissions, in one scenario Allen et al. assumed that 75% of all controllers were pneumatic-style controllers, of which 75% would be actively venting methane. This assumption results in an estimated average of 1.5 venting pneumatic controllers per well.”

<https://www.epa.gov/sites/production/files/2015-12/documents/ng-petro-inv-improvement-pneumatic-controllers-4-10-2015.pdf>

838: Composition data is also available from GHGRP. The GHGI predominantly (though indirectly) uses CH₄ concentration data from GHGRP (by using mass of CH₄ reported to GHGRP for many sources).

860: Specify which EPA program uses these 4 categories. The GHGI does not.

965: Is there variation in the wells per site in each of these categories? Does that impact the comparisons with site-level data?

1072: Were all petroleum systems CH₄ emissions included in the numerator or just a fraction that was attributed to gas production versus oil production? Another option is to do a BOE comparison.

1083: This is true only for equipment leaks. For sources calculated with more recent data (basically all here except equipment leaks) those are considered net emissions.

1090: The first emissions year of GHGRP reporting for natural gas and petroleum was 2011 (reported in 2012), not 2009. The GHGI began incorporating those data in 2014.

1102: Note that the current GHGI (using GHGRP) values for year 2012 are still consistent with the Allen et al values measured in 2012. As more recent data from GHGRP continue to become available, they (for the most part) show a decreasing trend from 2012.

1132: Please recheck the references. The GRI study was commissioned by GRI and EPA, and conducted (in part at least) by Radian. Unclear about STAR involvement.

1145: It might be helpful to label here which references are used in oil versus gas.

1173: Differences in the regional breakout of well locations and regional CH₄ content is likely what drives the differences.

1215: Are there potentially issues with the correlation equations? Are there recommendations for improvements?

1362: This is a key caveat. There is a chance the difference is due to the studies representing different regions, types of systems, etc. This should be clearer in the main body of the paper.

1394: Clarify which of the GHGI data sets are also included in this study.

1434: These are important points and should be covered in the main body

1456: Is this all of the equipment at the well or just wellheads?

1465: Is the GHGI value for # leakers missing here?

1672: The malfunctioning dump valves are reported for controlled and uncontrolled large tanks and are not reported for small tanks.

1680: What is it that's inherent to the simulation approach that results in an underestimate?

1686: Could another approach to address the gap be to include an improved EF for unintentional events? If that is where most of the difference is, this could be a more straightforward approach.

1933: There isn't much discussion in the paper about the split between eastern and western regions for equipment leaks in the GHGI, established in the 1992 GRI study. Is it possible that shifts in production and practices across the U.S. compared to 1992 also contribute to the gap?

Dear Reviewers,

Thank you for these detailed reviews.

Below we address each comment point-by-point in indented text, indicating any corresponding changes we made to the manuscript in double-indented text beginning with the term “Manuscript:”, “SI:”, or “Abstract:” as well as the specific location within the document of the update. We have re-numbered all references within this response document, including in excerpts from the text, to use a consistent scheme within the document.

Manuscript and Supplementary Information files are provided with “track changes”. Note that you will see two versions of the figures (as the strike out doesn’t appear in the PDFs). The first figure is the superseded version and second figure is the new version.

Regards,

Jeff Rutherford

Reviewer #1 (Remarks to the Author):

The authors construct a bottom-up emissions estimate and attempt to reconcile differences between EPA's greenhouse gas inventory with recent top-down studies that find much larger methane emissions from the oil and gas sector. I think this manuscript is well-written, the analysis is sound, and the article makes an important contribution to the field. I recommend the manuscript for publication. I have a few relatively minor comments and suggestions:

Line 80-81: Is there a reference for this statement? In theory, many atmospheric observations would reflect the integrated signal of emissions that occurred over many hours or a few days (depending upon the measurement).

Author: The reference at the end of the sentence is the source for this statement. We have moved the reference so the association with this statement is clearer.

Manuscript L87: Recent work suggests that top-down measurement campaigns are capturing systematically higher emissions during daytime hours from episodic events [1]. However, this may not be true at a national level, as it has been noted that the upward bias of top-down measurements was likely explained by unusually high liquids-unloadings in the Fayetteville shale [2].

Line 141: Why apply the bootstrapping algorithm 100 times? If the distribution of emissions from wells has a heavy tail and if you are particularly interested in extreme values, then it may be important to run more than 100 simulations to full capture that distribution. It's not uncommon to run 1e3 or 1e4 Monte Carlo simulations, depending upon the complexity of the distributions involved. Do you think that 100 simulations is sufficient?

Author: First, it should be noted that each Monte Carlo realization draws results for ~1 million wells individually. Because running a single instance of all ~1 million wells takes 2-8 hours, 100 bootstrapping iterations was the maximum allowable for this manuscript. However, most of the variability in emissions is already captured within each run through the modelling of ~1 million wells. This manuscript is part of a larger project of improving methane modelling in the Oil Production and Greenhouse Gas Emissions Estimator (OPGEE). OPGEE is a Microsoft Excel-based model which has slow run-times.

We have added a new section to the supplementary information illustrating convergence of our model with 100 Monte Carlo realizations.

SI Section 5.4.4:

This manuscript is part of a larger project of improving CH₄ modelling in the Oil Production and Greenhouse Gas Emissions Estimator (OPGEE). OPGEE is a Microsoft Excel-based model which has slow run times. Because running a single instance of all ~1 million wells takes multiple hours, 100 bootstrapping iterations was the maximum allowable time for this manuscript. Further, in order to achieve 100 bootstrapping (or Monte Carlo) realizations in a reasonable time, we ran OPGEE in parallel on eight different computers (**Table S22**). Each computer received the same OPGEE file. The only difference is that each file is programmed to receive as inputs different batches of emission factor distributions (see description of distributions in Section 5.4.1).

Table S22: Descriptions of eight computers running OPGEE uncertainty realizations in parallel.

Computer #	Operating System	Excel version no.	Run #1	Run #2	Total Runs
1	MS Windows 10 Education	MS Office Professional Plus 2016	1-44	92-24	47
2	MS Windows 10 Education	MS Office Professional Plus 2019, v. 1808	45-56	73-74	14
3	MS Windows 10 Education	MS Office Professional Plus 2016	57-68	75	13
4	MS Windows 10 Education	MS Office Professional Plus 2016	69-72		4
5	MS Windows 10 Education	MS Office Professional Plus 2016	76-80		5
6	MS Windows 10 Education	MS Office Professional Plus 2016	81-89	95-96	11
7	Windows 10 Home, v. 1909	MS Excel for MS 365, Version 16.0	90-91		2
8	MS Windows 10 Education	MS Office Professional Plus 2016	97-100		4

In **Figure 1** and **Figure 2** (main text), we calculate error empirically (95% confidence interval based on the 2.5 and 97.5 percentile values extracted from the empirical distributions).

Here, we will use a different approach to illustrate model convergence with increasing Monte Carlo realizations. For this calculation, we estimate a 90% confidence interval (CI) using the formula:

$$\bar{\mu} \mp 1.645 \frac{\bar{\sigma}}{\sqrt{n}}$$

Where $\bar{\mu}$ and $\bar{\sigma}$ are the sample mean and standard deviation, which we expect will converge towards the expected values of μ and σ , at a rate of $1/\sqrt{n}$, with sufficient model realizations n . This formula assumes that the distribution of error converges towards a normal distribution.

After 100 Monte Carlo realizations, total CH₄ emissions results converge on an error of approximately 1% (with 90% probability, see **Figure S11**).

Figure S11: Evolution of (a) sample mean ($\bar{\mu}$) and (b) 90% confidence interval for $\bar{\mu}$ versus number of Monte Carlo realizations

Line 222: Are there studies of facility-level or component-level measurements that reached parallel conclusions?

Author: Yes, there are several references supporting this statement. We added references to Allen et al 2015 ([3], who perform direct measurements of liquids unloadings and find close agreement with the GHGI), Allen et al 2015b ([4], who perform direct measurements of pneumatic controllers and find close agreement with the 2015 GHGI, but which are low compared to the 2020 GHGI), and Allen et al 2013 ([5], who suggested a downward revision was required for completions and workovers). We also reference SI Section 6.1 where these studies, and any subsequent revisions to the GHGI are discussed in greater detail.

Manuscript L271: Results from our modelling (**Figure 3**), in addition to recent revisions by the GHGI and other analyses ([3]–[7], see further discussion in SI- 6.1), suggest that the downward bias of the GHGI is not primarily due to pneumatic devices, liquids unloadings, completions and workovers, methane slip from reciprocating engines, or uncombusted methane from flares (either the divergence is small, absolute emissions are small, or emissions are higher in the GHGI compared to our study).

Fig 4: What do the vertical maroon bars represent? These bars are quite large and encompass the EPA estimate. Hence, I think it would be a useful point to explain to the reader.

Author: The vertical bars represent the 95% confidence interval. This has been clarified in the figure caption.

Manuscript L360: Example decomposition of the equipment-level emission factor (and 95% confidence interval, represented by vertical bars) for Western US gas wellheads (Note that units differ for each panel, and also the logarithmic scale meaning that visible differences between points often span orders of magnitude).

Line 375: I think the quotation mark is in the wrong place.

Author: Fixed

Manuscript L463: Our detailed decomposition identifies (i) underlying equipment-leak measurements and (ii) neglect of the contribution of unintentional emissions events at tanks (e.g., liquid hydrocarbon tank “thief hatches”) as likely the most important contributors to the underestimation.

Line 394: Can you clarify what you mean by different spatial scales? Are you referring to atmospheric measurements and satellite measurements (i.e., TROPOMI), facility-level measurements, component-level measurements, or something else?

Author: This has been clarified in the text

Manuscript L506: Regular efforts to validate equipment-level emission factors by comparing existing or new emission factors with measurements from randomly sampled sources at different spatial scales (i.e., validating component-level, direct measurement campaigns with downwind truck or airplane-based measurements) would also improve accuracy and “build in” to inventory efforts the ability to correct data over time.

Reviewer #2 (Remarks to the Author):

This is a very thorough and novel effort. The major claims of the paper are that the observed gap between top down studies and the U.S. GHG Inventory is not due to an inherent issue with bottom-up approaches, that the gap can be considered to be primarily due to underestimates in equipment leak and tank emissions, and that improved modeling of equipment leak and tank emissions can address the gap. These are novel claims and will be of great interest to the community. The paper will likely influence thinking and potentially, future research efforts. Because of the potential influence, it is important that additional clarification and information is provided to either strengthen the conclusions or enhance transparency about caveats and potential alternative explanations for the gap.

The authors have conducted a comprehensive review of available data and compiled a very useful database. Data in both the GHGI and the alternate data sets used in this study are somewhat limited (as only limited data exist). Use of the alternate data sets addresses the gap, but if those data sets themselves are inaccurate or not representative of national emissions, then this paper could be incorrectly eliminating other potential sources of the gap, and influencing research efforts to move away from those and towards leaks and tanks.

For example, for tanks, the key data set with year 2011 measurement data incorporated into the model is limited to one region. It is unknown how representative 2011 tank emissions in that region are of tank emissions in the U.S., and therefore how informative that data set is to understanding national-level gap.

Author: The issue of data representativeness is important, and we agree this should receive more attention in our manuscript. We more clearly acknowledge this as a limitation of our study in the abstract and introduction:

Abstract L28: We rely on the most comprehensive and up-to-date set of component-level quantified emissions measurements available, although even these have substantial room for improvement.

Manuscript L119: We acknowledge that the results of our study required extrapolating relatively small sample sizes to the level of the US. Certain sources, especially tanks, are currently poorly characterized, and this prevents us from generating region-specific emission factor estimates. However, when evaluating our results, we must be clear that the baseline we are comparing to is not a world with perfect information about CH₄ emissions. It is the current GHGI, which is even more data limited.

However, in addition to validating our model on the basis of total CH₄ emissions, we have performed other validation exercises against site-level studies. These validation exercises, now summarized in SI-5.5, demonstrate how our model captures mechanistic trends observed in site-level studies (for example, the relationship between production normalized emissions and site-level gas productivity). We also note that variability introduced by operator practices and state-level regulations will not be captured by our model. Especially given the severe component-level data limitations.

This is now summarized in the main text, both in the early methods description and in our discussion:

Manuscript L164: As both top-down and site-level measurement studies have demonstrated, there is a wide variability in CH₄ emissions across O&NG production regions [2], [8]. Some of this variability will be captured through data sources and mechanics of our model, and some will not. As Omara et al. [9] demonstrate, a significant share of this variability can be explained by the combination of number of sites and natural gas production characteristics. Our model is able to replicate Omara et al.'s relationship between site-level productivity (Mscf/site/day) and production normalized CH₄ (i.e., basins with low productivity sites demonstrate higher production normalized CH₄, [9] see SI 5-5 and **Figure S12**). We are also able to demonstrate a second trend from the site-level literature (e.g., [10], [11]) where emissions per site are higher at liquids-rich sites versus gas-rich sites (**Figure S13**, noting however that this trend is weak, and should only be considered suggestive). While we believe, based on these validation exercises that our model can describe variability across basins relatively well, we acknowledge that our results are still constrained by the limited number of component-level measurement studies available. Beyond the production-related factors described above, variability will also be introduced by regulatory frameworks and

operator practices that differ between regions. If data were available as a representative sampling of component-level measurements across basins, our method could capture this variability. However, given the data limitations, our measurements are biased towards certain geographies (e.g., tank measurements are sourced entirely from the ERG 2011 Fort Worth campaign [12]). As measurement campaigns progress over time, this issue should diminish.

Manuscript L467: By collecting and synthesizing all available component-level measurement data into a singular database, we believe this study provides the clearest assessment of CH₄ emissions from the US O&NG production segment. Pooling of studies was necessary, given that research on superemitters to date has demonstrated that “larger sample sizes are required ... to achieve targeted confidence intervals” [13]. However, as we have described, our data may not adequately represent all regions of the US, especially for certain source categories. Sub-sampling in our larger dataset to focus on particular regions or types of facilities may offer spurious improvement, wherein specificity for that region or type of facility may be improved, but generalizability is hindered because the sample sizes for each new sub-sample become small. Future research should target data collection to fill these gaps in the literature to improve size and representativeness of samples.

Author: There is also a new section of the supplementary information dedicated to this issue.

SI – 5.5:

Dozens of distinct O&NG producing basins are scattered across the US. Each of these regions will have different geological and operations related characteristics affecting CH₄ emissions. Some of these characteristics are addressed by our model, and some are not, suggesting the need for further research. In this section, we will begin by discussing the aspects that are captured by our model. We will conclude by addressing possible bias in what is not captured by our model.

The relationship between site-level productivity (Mscf/site/day) and production normalized CH₄ emissions has been discussed extensively in Omara et al. [8]. According to Omara, top-down studies have demonstrated a wide variability in production normalized CH₄ emissions across basins, and “these trends are caused by differences in the distribution of both the number of sites and their natural gas production characteristics”. Basins such as Appalachia, Greater Green River, and Arkoma produce most of the gas from highly productive sites (>70% of gas from sites > 1,000 Mscf/site/day). Based on Omara’s model, which is calibrated to downwind, site-level surveys (see Section 2), these sites also demonstrate low production normalized CH₄ emissions (<1.5%). It is the opposite case for low productivity basins such as San Juan and San Joaquin (10% and 7% of gas, respectively, from sites > 1,000 Mscf/site/day) which demonstrate high production normalized CH₄ emissions (>4%). In **Figure S12**, we demonstrate how our model reproduces the same relationship as Omara between site level productivity and production normalized CH₄ emissions.

Figure S12: Relationship between site-level productivity and production normalized emissions (e.g., fractional loss rate) for this study and Omara et al. [8], respectively (model fits calculated using a quadratic weighted least squares regression).

Site-level studies have also demonstrated higher emissions at liquids-rich sites versus gas-rich sites [11], [14]. This is likely due to a higher prevalence of high-emitting equipment like tanks at liquids-rich sites, which is reflected the activity factor parameterization of our model (see **Table S13**). We note, however, that this relationship between liquids production and site-level emission in our model is relatively weak, and only provides suggestive evidence (emission per site are only ~10% higher at liquids-rich sites, **Figure S13**).

Thus, our model demonstrates key mechanistic trends which have been shown to explain a substantial portion of variability in CH₄ emissions across US basins. However, unlike the site-level data from Omara et al., few component-level measurement studies provide information on gas and oil production volumes of measured wells and equipment. Our model can replicate Omara’s “scale dependence” through application of a “conservation of mass”, described in detail in Section 4.1.

Figure S13: Probability distribution of emissions rate per site for subsets of results distinguished by production stream. Note that “oil + gas” refers to sites with a GOR < 100 mscf/bbl producing associated gas, and “gas + oil” refers to sites with a GOR > 100 mscf/bbl producing associated liquids.

Omara et al. [8] also acknowledge that “other factors ... such as new state/local regulations or voluntary emissions reduction programs performed by specific operators likely also contribute to basin-to-basin variability”. These factors would differ across operators, jurisdictions, and time periods. With a geographically representative set of component-level CH₄ emissions measurements, hypothetically, it would be possible for our model to capture this variability. However, although we include measurements taken from most major gas producing basins, our limited data set does not mirror exactly the distribution of US gas production. We acknowledge this is likely a source of bias in our model. For example, a large proportion of quantified emissions measurements for storage tanks, and to a lesser extent equipment leaks, are derived from ERG’s [12] Fort Worth campaign. Further, our data sample includes measurements published between the period 1993 – 2019. Clearly operating practices have changed

during this time period, therefore older data may be less representative of conditions in 2015 (our basis).

Figure S14: Quantified measurement count by study for key emission sources. The legend is annotated to indicate study measurement locations.

When evaluating the utility of our results, we must be clear that the baseline we are comparing to is not a world with perfect information about CH₄ emissions. It is the current GHGI, which is even more data limited. As we elaborate in Section 6.2, the GHGI emission factors for equipment leaks are based on a regional weighting of “Eastern” and “Western” CH₄ emissions data. However, by regionally segmenting data, this has resulted in very low sample sizes by component for Eastern US gas (e.g., 10 quantified measurements of open-ended lines and 24 measurements of valves). Therefore, while component-level emission factors in the Eastern data (e.g., **Figure S19**) are significantly smaller compared to both this study and the EPA Western US data, it is unclear if this is based on significant differences in operations. Since these measurements were made, NG production in the Eastern US has grown from <5% of US domestic production to ~28% (**Figure S14**).

Figure S14: Comparison of gas production, wells, and estimated share of US production sector CH₄ emissions across oil producing wells, and NG producing wells in the Western and Eastern US. This differentiation matches the different data sources used in the GHGI.

Reviewer: For equipment leaks, some of the key input data for the equipment leak model come from studies that calculated lower leak emissions per equipment than the GHGI (Allen et al. and Pasci et al.) and some of the input data come from the same studies used to develop the GHGI (STAR). The authors could provide additional information on how the deconstructed data from those studies resulted in higher emissions than the GHGI in their model.

Author: As the reviewer points out, some of the cited studies calculate lower equipment-level emission factors and total equipment leak emissions compared to both our study and the GHGI. However, our study does not take the equipment-level emission factors as inputs. Rather, we take the combined component-level emission data, component counts, and fraction of components found to be leaking.

For example, Allen et al [5] document leaking components only and scale emissions using well counts. Because a full component count was not reported, we are unable to calculate a “fraction leaking” statistic for comparison with our study and the GHGI. Pasci et al. [15] do apply a complete component count. Component-level emission factors are higher compared to API [16], but the “fraction leaking” is lower, leading to slightly lower population emission factors.

We have added a brief statement addressing this issue in the main text:

Manuscript L146: Note that some of the cited studies will also calculate equipment-level emission factors. However, our study does not take the equipment-level emission factors as inputs. Rather, we take the combined component-level emission data, component counts, and fraction of components found to be leaking, therefore values calculated here will be different from the values calculated in those studies.

It is important to note that the fraction leaking values of Pacsi et al. [15] are near the lower end of our sampled range. However, a more important reason why our study still finds higher emissions compared to the GHGI is that the GHGI applies a weighted average of Western and Eastern emission factors and Pacsi et al [15] only compare with Western factors. The Eastern factors, which now represent a substantial fraction of gas wells in the GHGI due to the shale boom, are significantly smaller compared to Western factors.

We have concerns with the Eastern factors. This is now addressed in the manuscript:

Manuscript L346: One source of the difference not illustrated in **Figure 4** between our study and the GHGI is related to how equipment-level emission factors in the GHGI (for NG systems) are a region-weighted combination of Western US and Eastern US factors. Component-level emission factors in the Eastern data (e.g., **Figure S19**) are significantly smaller compared to both this study and the EPA Western US data and are derived from an even smaller sample from the 1990s (~100 quantified leaks). Since these measurements were made, NG production in the Eastern US has grown from <5% to ~28% of total US production (**Figure S14**).

Reviewer: The other input data for equipment leaks come from a Canadian study, which the authors note may be impacted by different regulations/practices. Information on any QC performed on the model would also be helpful.

Author: A goal of this study was to use exclusively US data to the greatest extent possible. We were able to accomplish this objective for emissions data, however, it was necessary to use the Clearstone [17] data for component counts. We now make this point in the supplementary information where the reviewer has suggested a line edit:

SI L453: Due to severe data limitations, note that we make an exception to our filters and apply Canadian data from Clearstone [17] in component counts and fraction of components emitting (for regulators only). No Canadian datasets are used for emissions data. We would expect component counts per equipment to be to a large extent standardized between US and Canada.

Reviewer: It is unclear from the information presented to what extent the gap is addressed through use of different input data versus through a different modeling approach. More information on how the study input data (e.g. a table with average EFs) compare with the GHGI will help strengthen the argument that it is the modeling approach that is addressing the gap versus different emission factor data.

Author: It is a combination of factors which include both data (various parameters including component-level emission factors and component-level counts per equipment) and modelling decisions (e.g., simulated versus measured emissions for storage tanks) that lead to our model estimating higher emissions compared to the GHGI. The reviewer has indicated (in line edits below) several places in the text where this could be clearer. We have modified our discussion accordingly.

Manuscript L478: These results demonstrate that the bottom-up methodology is a valid approach to produce accurate emissions estimates and that improvements to inventory

methods are possible through both more comprehensive datasets and revised modelling approaches (demonstrated through respective contributions to the decompositions in **Figures 4** and **5**).

Manuscript L528: We believe that incorporation of a larger emissions dataset and revised modelling approaches to sources including storage tanks and flaring has produced a more accurate inventory estimate for production segment CH₄.

Reviewer: The paper considers the gap to be closed based on developing similar emissions estimates using component level to those developed with site-level data. Additional information about the site level study (e.g. how many sites were measured by the site level campaigns, how much variation there is in those measurements, were the sites oil versus gas, high or low production, geographic scope) and how the comparison was done (e.g. be clearer that this study did not develop estimates for the specific sites measured in the site-level campaign) would strengthen the paper.

Author: We have added further description of the Omara and Alvarez studies to Section 2 of the supplementary information.

SI Section 2

We validate our component-level bottom-up model by comparing total CH₄ emissions and emissions distributions with those generated in site-level studies (Alvarez et al. [2] and Omara et al. [8], respectively). Details of how results were prepared for this comparison are given in Section 5.3. Here, we will briefly summarize the studies compared.

Site-level studies rely on vehicles or other mobile laboratories equipped with CH₄ sensors downwind of well-pads. As the name implies, resolution is limited to aggregate emissions at the scale of a well-pad or processing facility. Methods include:

- *EPA Other Test Method (OTM 33A)*: An EPA OTM 33A assessment begins with an initial drive-by screening where the laboratory equipped vehicle searches for elevated CH₄ measurements. This screening is followed by a continuous measurement of CH₄ where the truck is parked downwind of the CH₄ emissions sources. Emissions are then estimated using a Gaussian approach. Examples of OTM 33A studies include Brantley et al. [18] and Roberson et al. [19].
- *Downwind tracer flux (DT)*: In a downwind tracer flux assessment, an atmospheric tracer (e.g., acetylene and nitrous oxide) is released near an emissions source and the mobile laboratory is driven downwind. Detailed post-hoc modelling is not required because it is assumed that dispersion of the tracer and CH₄ is similar. Examples include Goetz et al. [20] and Omara et al. [21].
- *Downwind measurements with Gaussian dispersion modelling (MMG)*: Some studies (e.g., Yacovitch et al. [22]) describe their approach as downwind measurements with Gaussian dispersion modelling. This approach is like the EPA Other Test Method 33A, with the notable exception that measurements can be taken while the vehicle is moving.

Omara et al. [8] synthesized results of eight production-segment studies: Brantley et al. [18], Eastern Research Group [12], Goetz et al. [20], Lan et al. [23], Omara et al. [21], Rella et al. [24], Robertson et al. [19], and Yacovitch et al. [22]. Some new measurements were also added. In total, the dataset includes site-level CH₄ emission measurements across 1009 sites (1 site = 1 measurement) and eight basins. Sampled basins include both gas producing regions (Marcellus, Fayetteville) and oil producing regions (Denver Julesburg, Eagle Ford).

The aim of the Omara study was to assess: (i) total CH₄ emissions from natural-gas production sites, (ii) basin-to-basin differences in CH₄ emissions, and (iii) the relationship between CH₄ emissions and natural gas production. Omara et al. constructed a national-level production-segment CH₄ emissions estimate by using a non-parametric bootstrap resampling approach. Before sampling, emissions measurements were grouped into ten bins based on deciles of natural gas production (Noting that Omara et al. do not estimate CH₄ emissions for wells reporting zero natural gas production, whereas Alvarez et al. do). The 498,000 US natural gas production sites were binned into the same ten bins.

Note that the distribution of sampled sites in Omara et al. are generally higher production compared to the overall population of US sites (Fig S21 in Omara). Although this is accounted for in the model, low producing sites may not be characterized as well as high producing sites.

Table S1: Summary of site-level data sets used in Omara et al and Alvarez et al (based upon statistics reported in Table S8 of Omara et al. [8]). Methods include EPA Other Test Method (OTM), downwind tracer flux (DT), and mobile monitoring with Gaussian modelling (MMG).

Study	Region	Omara	Alvarez	Number of sites	Measurement Technique	Site-level gas production (Mcf/d), Min-Max
Omara et al. [21]	Marcellus	Y	Y	18	DT	0.68—44
Omara et al. [21]	Marcellus	Y	Y	13	DT	450—78,000
Brantley et al. [18]	Barnett	Y	Y	43	OTM	3.7—5,160
Brantley et al. [18]	Denver Julesberg	Y	Y	74	OTM	4.9—1,830
Brantley et al. [18]	Eagle Ford	Y	Y	4	OTM	78—2,000
Brantley et al. [18]	Pinedale	Y	Y	106	OTM	4.6—9,000
Rella et al. [24]	Barnett	Y	Y	185	MMG	2.3—6,000
ERG [12]	Fort Worth	Y		287	Onsite	0.4—39,300
Omara et al. [8]	Denver Julesberg	Y		18	DT, OTM, MMG	1—5,470
Omara et al. [8]	Marcellus	Y		45	DT, MMG	40—25,200
Omara et al. [8]	Uinta	Y		29	DT, OTM, MMG	4—3,580
Goetz et al. [20]	Marcellus	Y		3	DT	4670—8,360
Lan et al. [23]	Barnett	Y		32	MMG	4.5—4,150
Yacovitch et al. [22]	Barnet	Y		7	MMG	23—1,160
Robertson et al. [19]	Fayetteville	Y	Y	50	OTM	21—4,350
Robertson et al. [19]	Uinta	Y	Y	29	OTM	2.2—1,160
Robertson et al. [19]	Denver Julesberg	Y	Y	15	OTM	2.1—326
Robertson et al. [19]	Green River Basin	Y	Y	51	OTM	7.1—3,380

Similarly, Alvarez et al. [2] also extrapolate a sample of site-level studies to produce an estimate of US CH₄ emissions for the production segment. Alvarez rely upon a smaller sample of studies: Brantley et al. [18], Omara et al. [21], Rella et al. [24], and Robertson et al. [19]. The extrapolation approach is also different. Rather than bootstrapping, Alvarez et al. derive probability density functions of emissions assuming a power law relationship between emissions and gas production. The Alvarez et al. model assumes that the underlying emissions distributions are lognormal.

Reviewer: It would be helpful to provide clearer information on the relative contribution of petroleum production versus natural gas production for the gaps in tank and leak emissions. It appears that the tank difference is largest for petroleum systems (model is around 1,400 Gg larger than GHGI for oil, and around 600 Gg larger than GHGI for gas), and that for leaks its larger for gas (1,000 Gg larger than GHGI for gas, and 500 Gg larger for oil). Also, my sense is that emissions data from oil production sites is far

more limited than from natural gas production sites. If this is the case, the paper should be clearer about this—are the conclusions on the sources of the gap stronger for natural gas than for petroleum?

Author: We have added statements to both the equipment leaks and storage tanks sections quoting the difference between this study and the GHGI for petroleum versus natural gas systems. Both emission factors and total emissions are listed in Tables S3 and S4 so we have added references to the main text.

Manuscript L354: Equipment-level emission factors and total emissions for each equipment class are also presented in **Table S3** and **Table S4**. Taken together, the gap between this study and the GHGI for equipment leaks is higher for natural gas systems (1.0 Tg) versus petroleum systems (0.5 Tg).

Manuscript L440: Equipment-level emission factors and total emissions for intentional flash emissions and unintentional emissions are also presented in **Table S3** and **Table S4**. The gap between this study and the GHGI is much higher for petroleum systems (1.4 Tg) versus natural gas systems (0.6 Tg).

Reviewer: The authors recommend applying a similar method to that used in this study to develop the U.S. GHGI and also GHGIs internationally. Additional rationale for this recommendation should be provided—given the uncertainties and the limited input data, what are the benefits of applying such a method to develop an annual GHGI versus instead applying EFs developed from such an analysis (or instead just applying average EFs from input studies)?

Author: We have clarified our recommendations by distinguishing those relevant to the US and those relevant to other countries. Because our data is specific to the US the emission factors can be used as-is, as the reviewer points out. For other countries, our results are relevant in two ways. First, as a possible replacement for the IPCC default Tier 1 emission factors. Second, as a general methodology for countries who wish to develop country specific Tier 2 emission factors.

Manuscript L501: Because all emissions data and activity factors (with some exceptions, noted in methods) are US-based, emission factors from this study (summarized in **Tables S2-S4**) could be implemented in US inventories. Emission factors for equipment leaks could be implemented relatively easily by updating existing sources categories. Implementing emission factors from storage tanks based on this study would require modifications to source categorization, for example, through the addition of a new factor to take into account failed controls like open thief hatches. Regular efforts to validate equipment-level emission factors by comparing existing or new emission factors with measurements from randomly sampled sources at different spatial scales (i.e., validating component-level, direct measurement campaigns with downwind truck or airplane-based measurements) would also improve accuracy and “build in” to inventory efforts the ability to correct data over time.

The results of this study are also relevant globally, both as inputs to default emission factor databases and as a generalized methodology for generating emission factors in different countries. All parties to the UNFCCC submit annual inventories, generated using a bottom-up approach, to report on progress towards GHG targets. The IPCC’s

Guidelines for National Greenhouse Gas Inventories outlines three approaches towards producing an inventory, with the simplest approach (Tier 1) based on IPCC default emission factors [25], [26]. Default emission factors for the petroleum and natural gas systems production-segment are based upon the same underlying data sets as the GHGI [26]. This means that, in addition to the US-submitted GHGI, other countries using Tier 1 emission factors will be contributing CH₄ estimates according to data that we have found likely to be underestimating of actual emissions. Recommendations offered herein, if implemented, may improve emissions estimates globally. Given the sparsity of data globally, we are unable to state how much error is introduced by use of these factors globally.

It should be noted, however, that at the time of writing of this publication IPCC Tier 1 emission factors are unlikely to be updated soon. For agencies wishing to improve the accuracy of Tier 2 emission factors this study identifies sources towards which efforts should be focused (some countries, e.g., Canada and Australia [17], [27], have requisite component-level data). We believe that incorporation of a larger emissions dataset and revised modelling approaches to sources including storage tanks and flaring has produced a more accurate inventory estimate for production segment CH₄. Finally, although our focus in this paper is on inventory development, the results of this study will also be relevant to industry in targeting and prioritizing practices to reduce CH₄ emissions.

Reviewer: Finally, the authors have developed a unique understanding of the available data that can be used to develop GHG Inventories, and how limited that data can be. Many research efforts on methane do not produce data that can be used in the type of inventory analysis conducted by the authors or do not focus on emission sources most likely to contribute to the gap. If appropriate, recommendations on future research priority areas based on the authors' experience could enhance the influence of the paper.

Author: Based on the reviewer's earlier comments, and some additional line edit suggestions, we now emphasize in the main article's conclusions the need for more component-level data collection. Emphasis especially is given to tanks.

Manuscript L425: Although the ERG study benefited from unique site access granted by municipal authorities, future studies should prioritize access to tank walkways and consider pursuing additional measures to sample thief hatches, pressure-relief valves, and vent stacks (ERG document the use of extensions to the High Flow Sampler tubing to access out-of-reach components and large nylon bags to sample oversized openings such as thief hatches [12], [28]).

Manuscript L467: By collecting and synthesizing all available component-level measurement data into a singular database, we believe this study provides the clearest assessment of CH₄ emissions from the US O&NG production segment. Pooling of studies was necessary, given that research on superemitters to date has demonstrated that "larger sample sizes are required ... to achieve targeted confidence intervals" [13]. However, as we have described, our data may not adequately represent all regions of the US, especially for certain source categories. Sub-sampling in our larger

dataset to focus on particular regions or types of facilities may offer spurious improvement, wherein specificity for that region or type of facility may be improved, but generalizability is hindered because the sample sizes for each new sub-sample become small. Future research should target data collection to fill these gaps in the literature to improve size and representativeness of samples.

Manuscript L492: For development of emission factors for crude and condensate storage tanks, differences are believed to be largely a result of the GHGI neglecting emissions from failed tank controls (e.g., open thief hatches). Although we attempt to estimate their contribution, and reference supporting site-level surveys, tank emissions remain a significant data gap.

Line by line comments:

Main text:

103: Explain the relative contribution of each (improved dataset and different statistical methods) for closing the gap.

Author: We consider this a result rather than an introduction element. As we note in response to an earlier comment, we have added text at main manuscript locations L478 and L528.

104-105: The comparison between site-level results and the study's results could be explained more clearly. My understanding is that there was no modeling done in this study for the sites that were measured for comparison so it's more that a modeling of national level emissions using site-level measurements was compared with a modeling of national level emissions using component-level measurements.

Author: The wording has been clarified.

Manuscript L110: As mentioned earlier, site-level synthesis studies have been validated against even larger-scale top-down studies, so improved alignment between the national results of our component-level method and previous site-level synthesis results suggests much better agreement with top-down results [2], [9].

123: Specify the timeframe over which these data were collected.

Author: This is now acknowledged in SI Section 5.5 (See also response to general comments above).

SI L1448: Further, our data sample includes measurements published between the period 1993 – 2019. Clearly operating practices have changed during this time period, therefore older data may be less representative of conditions in 2015 (our basis).

129: Suggest providing information here the representativeness of the component-level EF dataset. For instance, are the data from different regions and different time periods?

Author: See discussion of data representativeness in a comment above (referencing manuscript L119, L164, and L467).

188: Is this a recommendation for how to compile the GHGI or a recommendation for how to develop EFs for GHGIs?

Author: We have revised the text to refer specifically to emission factors. As we discuss in the next paragraph, activity factors are identical to the GHGI.

Manuscript L233: Our results show that updated emission factors, through both more comprehensive datasets and revised modelling approaches, can recreate observed super-emitters.

242: These distinctions are made in the GHGI calculations. EPA provides results on its website at a higher-level but can also provide the input information.

Author: This paragraph has been edited so it no longer implies that this regionalization is no longer used, when it in fact is just not represented in the public inventory.

Manuscript L293: The GHGI further segments emission factors beyond petroleum and natural gas systems. Consistent with the underlying studies from the 1990s [16], [29], GHGI equipment-level, equipment leakage emission factors for natural gas systems are subdivided by region (Western gas versus Eastern gas), and for petroleum systems data are subdivided by product stream (light oil versus heavy oil). Equipment-level emission factors for gas systems, for example, are a weighted average of both Western emission factors and Eastern emission factors. The GHGI approach to aggregating these factors to overall values for natural gas and petroleum systems is described in SI-6.2.

299: Is there a recommendation for how to incorporate these components into GHGI?

Author: After further consideration we have modified the statement to reflect that this is less of an issue with the GHGI (see adjustment to main text). Although the component classification scheme in the GHGI (based on GRI report, 2-4 components per equipment) is smaller compared to our study (12-fold classification) we do not think this is a critical issue worth highlighting, or making recommendations on in the main text (we also elaborate on this in the supplementary information).

Manuscript L375: Second, we also limit this figure to connectors, valve, and open-ended lines (which account for the majority of components although our inventory and the GHGI also account for pressure relief valves, compressor seals, and other components in smaller numbers).

SI L1760: We also note that some specificity in component-level data was lost when measurements from API 4589 [16] and Star Environmental [29] was applied to the GRI report [30] and API 4638 [31]. Although 9-component and 7-component classification schemes were used in API 4589 and Star Environmental, respectively (see detail in Section 7), this detail was lost when the data was applied in the GRI report and API 4638 (reduced to a 4-component classification scheme). This is likely because correlation equations and pegged source factors are only available for a limited number of components (see Table S23, Table S24). However, given that the majority of components

(and their emissions) for equipment leaks can be attributed to connectors, valves, and open-ended lines, this issue is not of primary concern.

313: Are these tank emission modeling programs based on direct measurements though? If so, is this then comparing one way of incorporating direct measurements into a model with another? Are there any readily identifiable specific issues with the models used in GHGRP?

Author: Yes, this is correct. GHGRP emission estimates for tanks are based upon measured temperatures, pressures, and the oil production rate at the separator.

Manuscript L391: First, whereas our model is based on direct measurements, the GHGI is based on operator reported simulations from software programs such as API E&P Tank or AspenTech HYSYS [32], [33] (or rather, simulated emissions which are a function of measured process parameters such as temperature and pressure, see 98.233(j) of [34]).

We have also made it clear in the supplementary information that simulated tank emissions (using programs E&P Tank and HYSIS) have been demonstrated to be poor predictors of emissions.

SI L2048: One argument in favor of our approach, versus simulations, is the generally poor correlations that have been demonstrated between E&P Tank and HYSIS software and measurements [35].

342: This limitation of the tank data is an important caveat that should be discussed in more detail. There is anecdotal and observational information and (limited) measurement data indicating that the GHGI underestimates tank emissions. However, closing the gap for tanks here seems to have been done by assuming that a limited number of measurements in a single region is correct (or at least more correct) and that the national level estimates based on facility-reported tank modeling is incorrect. Based on available data, the approach makes sense, but the authors should be clearer that there are uncertainties here.

Author: See discussion of data representativeness above (referencing manuscript L119, L164, and L467). The most relevant sections are pasted below:

Manuscript L119: We acknowledge that the results of our study required extrapolating relatively small sample sizes to the level of the US. Certain sources, especially tanks, are currently poorly characterized, and this prevents us from generating region-specific emission factor estimates. However, when evaluating our results, we must be clear that the baseline we are comparing to is not a world with perfect information about CH₄ emissions. It is the current GHGI, which is even more data limited.

Manuscript L180: Beyond the production-related factors described above, variability will also be introduced by regulatory frameworks and operator practices that differ between regions. If data were available as a representative sampling of component-level measurements across basins, our method could capture this variability. However, given the data limitations, our measurements are biased towards certain geographies

(e.g., tank measurements are sourced entirely from the ERG 2011 Fort Worth campaign [12]). As measurement campaigns progress over time, this issue should diminish.

344: If possible/appropriate, build out recommendations here. It seems that truly closing the gap would require a lot of tank measurements, which are challenging. If the ERG study approach is a good model, note that.

Author: Agreed. We have clarified in the text, below, that aspects of the ERG sampling design could be difficult to replicate by others (such as unlimited site access granted by the City Gas Inspector). However, we also note aspects that could be applied more easily (such as extensions to the High Flow Sampler hose for hard-to-reach areas and nylon bags for sampling thief hatches).

Manuscript L425: Although the ERG study benefited from unique site access granted by municipal authorities, future studies should prioritize access to tank walkways and consider pursuing additional measures to sample thief hatches, pressure-relief valves, and vent stacks (ERG document the use of extensions to the High Flow Sampler tubing to access out-of-reach components and large nylon bags to sample oversized openings such as thief hatches [12], [28]).

SI L2497: The ERG study had unique advantages compared to other component-level campaigns. Given that the municipal government granted full site access to ERG teams (the City Gas Inspector, who always has access to well pads, was present for all site visits), results of the ERG study are likely less subject to sampling bias. Most studies rely upon voluntary study partners, potentially oversampling better-performing operators.

These unique access provisions provided additional benefits for emissions quantification. First, it is less likely that tank emissions were missed given that each IR camera survey involved “climbing up the stairs to the tank walkway in order to view each thief hatch and pressure relief valve (PRV) vent line” [12]. Other studies have noted difficulties measuring tanks due to safety issues [36]. If emissions were detected on tanks, remarkable efforts were made in quantification. For example, ERG document the use of extensions to the High Flow Sampler tubing to access out-of-reach components, large nylon bags to sample oversized openings such as thief hatches, and even the “use of a man lift and operator to provide the point source crew with access to the emission point” [12], [28]. Finally, beyond the unique access provisions of this study, emission sources were classified with a high degree of detail.

358: It’s surprising that the study has a different fraction for uncontrolled tanks than the GHGI (which uses GHGRP data). There is not another dataset for fraction of control with as much coverage as GHGRP. Please clarify the source of this fraction.

Author: Although both our study and the GHGI calculate fraction of tanks with controls using GHGRP data, we apply the data to our modelling differently. While OPGEE applies a universal value to all wells (both gas systems and oil systems), the GHGI applies separate values. This is why different values are shown in the figures.

We agree that this might be confusing. An explanation has been added to supplementary information where we discuss the decomposition plots for storage tanks.

SI L2288: Note that although both our study and the GHGI are based upon GHGRP data for fraction of tanks without controls (second panel in decompositions), OPGEE applies a universal value of 51%, whereas the GHGI applies differentiated values for oil versus gas systems.

357: In the Dallas ft worth study, storage tank breathing and standing losses were not measured/calculated. Non routine emissions also were not included. Does that mean that that study could be an underestimate itself? The study also used EPA correlation equation for tank calcs.

Author: We clarify, in the SI text, measurements of tanks that were made by ERG in the Fort Worth study. Although calculation of flashing, working, and breathing losses were not made, measurements of tank thief hatches, pressure-relief valves, and vents were made to the extent that they were identified by a FLIR camera or Method 21. We also clarify in the SI text how correlation equations are applied by ERG.

Note that we do account for flashing, working, and breathing emissions with calculations based on measurements made in the HARC report ([37], see description of these calculations in SI 4.3.2).

SI L2528: The ERG report documents significant emissions from tanks. For example, “the largest source of emissions detected with the IR camera was leaking tank thief hatches”. It is confusing, however, that the study notes that “other sources of emissions, including but not limited to, storage tank breathing and standing losses ... were not calculated. Non-routine emissions such as those generated during upsets or from maintenance, startup, and shutdown activities were also not measured or calculated as part of this study unless they were observed at the time of the site visit.” We assume this statement is intended to reflect that no additional calculations were made to reflect emissions resulting from the normal functioning of uncontrolled tanks (e.g., emissions released during the filling of tanks or due to diurnal changes in temperature). Estimates of tank flashing and working and breathing emissions requires measurement of tank vents over periods up to 24 hours (to capture a full diurnal cycle). However, measurements of tank thief hatches, pressure-relief valves, and vents were made with the High Flow sampler if they were identified by a FLIR camera or Method 21 screening.

ERG collected canister samples at a subset of well pads and laboratory analysis of the canister samples is used to speciate the emitted gas (the High Flow Sampler only reports emissions as a flow rate of natural gas). Because canister samples for speciation were not obtained at all facilities, correlation equations were developed based on the emissions results from canister measurements (in total organic carbon, lb/year) and the corresponding gas flow rate of the High Flow Sampler (reported %CFM). The methods applied are based on the guidance from the EPA’s Protocol document [38].

ERG also note that some low-level emitters were below the detection capability of the High flow sampler. For Phase II measurements, some of these low-level emissions were estimated using default zero factors from the EPA Protocol document [38].

368: Explain the validation in more detail. Were the specific sites measured by Omara et al modeled here?

Author: As noted in earlier comments, we have added further description of the Omara and Alvarez studies to Section 2 of the supplementary information. Omara and Alvarez use different modelling approaches to extrapolate a sample of facility-level measurements to the US total. We have also added a citation to Section 5.3 of the supplementary information, where the validation is discussed in more detail. Using data from Omara, we “cluster” outputs from our model, such that are outputs are on a facility-level basis, consistent with Omara (a production site will contain one or more wells and other associated production equipment).

We have added a reference to these sections at an earlier location in the paper.

Manuscript L202: We also validate our bottom-up tool by comparing total emissions and emissions distributions with those generated in site-level synthesis studies. The total CH₄ emissions estimate of our model is compared with Alvarez et al. [2], and site-level distributions are compared with Omara et al.[9] (see description of site-level studies in SI-2 and methodological elements of the validation exercise in SI-5.3).

375: Consider expressing as “as likely the most important contributors”

Author: We have modified the text.

Manuscript L463: Our detailed decomposition identifies (i) underlying equipment-leak measurements and (ii) neglect of the contribution of unintentional emissions events at tanks (e.g., liquid hydrocarbon tank “thief hatches”) as likely the most important contributors to the underestimation.

398: This line seems to be referring to the 2006 IPCC Guidelines for National Greenhouse Gas Inventories. The title should be corrected. Note that there are several versions of the IPCC guidance. The 2006 guidance likely (documentation not always clear) includes U.S. and Canadian data as input to default factors.

Author: The title has been corrected. We have also modified the citations to make it clear that we are referring emission factors from the 2019 refinement to the guidelines. In the 2019 refinement, EPA data is applied to all non-oil sands production segment emission factors and Canadian data is applied to oil-sands production segment emission factors.

Manuscript L515: The IPCC’s Guidelines for National Greenhouse Gas Inventories outlines three approaches towards producing an inventory, with the simplest approach (Tier 1) based on IPCC default emission factors [25], [26]. Default emission factors for the petroleum and natural gas systems production-segment are based upon the same underlying data sets as the GHGI [26].

404: The 2019 Refinement is a final document, and updates to the U.S. GHGI do not carry through to the IPCC guidance..

Author: The wording referenced here has been deleted.

405: Instead of the study influencing IPCC Tier 1 EFs, which is not possible in the near future since the guidance was most recently updated in 2019, the authors could instead note its relevance for countries wishing to develop country-specific Tier 2 factors.

Author: This issue is addressed in our general comments above (referencing main text L501).

406: Change text to UNFCCC inventory reporting. The methods come from IPCC; the reporting requirements (and specification of which version of IPCC GL to use) comes from UNFCCC. The authors could consider also noting that detailed emissions data may be helpful for countries with NDCs including oil and gas and this approach could contribute towards the development of robust source-specific estimates and track emissions over time.

Author: The wording referenced here has been deleted.

408: Provide a stronger argument that it's the approach versus the input data that's providing the improvement. Especially if the recommendation is meant to apply to other countries—data are extremely limited in other countries. Is a better recommendation to focus measurement efforts on leaks and tanks?

Author: As we note in response to an earlier comment, we have added text at main manuscript locations L478 and L528.

418: To be clearer and improve international relevance, it would be helpful to adopt IPCC terms (fugitives = emissions from leaks, venting and flaring).

Author: Since our model is applied to a direct comparison with the GHGI, we choose terminology consistent with the EPA (GHGI and GHGRP). Therefore, to avoid confusion, we have elected to not use the term “fugitives”. We have removed the word “fugitive” in reference to the OPGEE “VF subroutine” and replaced it with “CH₄ emissions subroutine”. This is clarified in the manuscript with elaboration in the supplementary information.

Manuscript L545: To avoid confusion, we do not use the term “fugitives”. To the extent possible, this study adopts the terminology conventions of the GHGI and the GHGRP with equipment leaks and vents (see further discussion in SI-1.1).

SI L61: Classification schemes for CH₄ emissions differ across regulatory contexts and jurisdictions. In its inventory guidelines, the IPCC refers to “fugitives” as comprising “venting, flaring, and leaks” [26]. However, in the U.S. the term “fugitives” is often used interchangeably with “leaks” in referencing unintentional emissions from components such as valves, connectors, or open-ended lines (for example, where the former term is used in “fugitive emission VOC standards” in Subpart OOOOa Section 60.5397a [39] and the latter terms is used as a general category of emissions in Subpart W Section 98.233(q) [34] and the GHGI). In both Subpart W and the GHGI venting emissions refer

to the intentional release of gas from equipment blowdowns, pneumatic devices, and storage tanks.

To avoid confusion, we do not use the term “fugitives”. To the extent possible, this study adopts the terminology conventions of the GHGI and the GHGRP with equipment leaks and vents. However, these terms should both be interpreted loosely. For example, our study integrates numerous data sources where quantified measurements are labelled by the source (e.g., valve or connector) but not by the purpose (i.e., intentional versus unintentional). Further, although the GHGI does not present additional categorization within oil and condensate tank vents (e.g., tanks with VRU versus tanks with flares, etc.), we differentiate between unintentional tank emissions and intentional tank emissions.

463: 51 is missing from the table

Author: This was an accidental reference to the ERG study. The citations have been fixed.

Manuscript L596: Data on fraction of components emitting was also scarce, with 3 studies containing useful information[12], [15], [16].

504: Enverus continually updates its well counts. It may be better to use latest Enverus counts for 2015. Another option is to add a sentence on how the total well counts in the Alvarez 2015 estimate compare with the 2015 well count estimate in the 2020 GHGI.

Author: The Enverus well dataset used in our study (based on Alvarez) is off by ~1.5% from EPA totals, which will not make a significant difference to overall results (and is within estimated uncertainty). We have added text to both the manuscript and supplementary information comparing the well counts totals between this study and the GHGI.

Manuscript L642: The total well count according to the Alvarez et al. Enverus dataset (1,005,191, see **Table S13**) is ~15,000 wells lower than the estimate of the EPA [40]. We discuss possible reasons for this difference (SI-5.1.2), but overall a difference of ~1.5% in well counts will not significantly affect our CH₄ emissions results.

SI L1068: Comparing well counts from the Alvarez Enverus dataset, our total count for onshore wells is ~15,000 wells lower compared to the EPA GHGI total (year 2015 data, as reported in 2020 inventory [40]). However, the EPA also uses the Enverus dataset for well counts [41]. In **Table S13**, we compare Enverus totals for onshore only with totals including offshore to see if this accounts for the difference, yet even including offshore wells there is still a ~10,000 difference between our totals and the EPA (note that this comparison is only for illustrative purposes, and offshore wells are not included in OPGEE modelling). Various filtering steps in Enverus could account for the differences. Also, Enverus is continually updating its database, and totals may differ depending on when data was downloaded. A difference of 15,000 is only 1.5% of total wells and will therefore not significantly affect emission results.

Table S13: Well counts for NG and petroleum systems compared with totals from the EPA GHGI [40]. Our study only models onshore wells in OPGEE, but we also present “onshore + offshore” totals for comparative purposes.

	Wells	Total prod. (MMbbls/year)	Total prod. (Bscfs/year)
Enverus database - Onshore only			
Gas	433,430	22	18,935
Oil	571,761	2,763	12,794
Total	1,005,191	2,786	31,729
Enverus database - Onshore + offshore			
Gas	433,881	24	19,334
Oil	576,511	3,367	13,851
Total	1,010,392	3,391	33,184
EPA GHGI			
Gas ¹	419,692		
Oil ²	600,519		
Total ^{3,4}	1,020,211	2,878	31,807

¹As reported in the row "Total Active Gas Wells" of Table 3.6-7

²As reported in the row "Total Oil Wells" of Table 3.5-5

³Total oil production reported as the activity factor for "Miscellaneous Production Flaring" on Table 3.5-5

⁴Total gas production reported as the activity factor for "Misc. Onshore Production Flaring" on Table 3.6-7

509: Provide additional explanation on how the authors were able to match EFs, component counts, and leak frequency to the different GOR levels, given how limited data are. Are data for certain GOR groups more limited than others?

Author: Note that we do not assign emissions data according to GOR or well productivity. We have added text to the manuscript so this is clearer.

Manuscript L651: When OPGEE iterates across each bin of wells, a conservation of mass (COM) conditional statement is implemented to ensure that the summed emissions do not exceed gas production (also accounting for the gathering and boosting, processing, transmission, and distribution sectors, see description of macro in SI-4.1.2). Note that the COM check is required because, unlike the site-level data from Omara et al. [8], few component-level measurement studies provide well-level meta-data (e.g., well liquid and gas production, well age, etc.) attached to emission measurements. Therefore, although well characteristics are binned for OPGEE, each bin draws upon the same sample set of emission measurements. Thus, in some instances, OPGEE can draw a leak that is larger than the volume produced, violating COM. These draws are rejected and redrawn to ensure COM.

We also elaborate on this in the supplementary information (although this is not new text).

SI L397: The second conditional statement is a conservation of mass (COM) check. At every well, a check is performed to ensure COM is not violated. In the COM check, the summed leaks are compared to total gas production at the well-pad. By applying the COM check we ensure total CH₄ emissions does not exceed 100% of gas (CH₄) productivity (also accounting for the gathering and boosting, processing, transmission, and distribution sectors). The only emissions source that is not included in the COM is completions and workovers since these emissions are infrequent and do not occur while the well is producing. If emissions exceed well-level CH₄ production, the macro loops back and iterates again for that well.

The COM check wouldn't be required if quantified emissions measurements from the literature were coupled with information on the gas throughput of the component. However, in contrast to the "site-level" studies synthesized by Omara et al [9], few (if any) "component-level" measurement studies provide information on the gas production volumes of measured wells and well-pad equipment. That is, these studies typically sample and quantify CH₄ emissions volumes from different pieces of equipment. Due to the missing volumetric production data, we only have CH₄ emissions volumes in absolute terms (e.g., scf per hour or kg per day), not in percentage loss terms. The application of COM to the iterative draws is what restricts high emitters at low productivity well-pads and is part of what leads to the "scale dependence" of fractional CH₄ emissions rates.

510: Provide additional explanation on the 60 bins of data, given how limited data are.

Author: I believe this is addressed with the comment above.

697: This was probably meant to be a reference to the Energy volume of the 2019 Refinement (not the forestry volume).

Author: The citation has been updated to reference the correct volume.

Manuscript L867:

E. Buendia *et al.*, "Volume 2, Chapter 4: Fugitive Emissions - 2019 Refinement to the 2006 IPCC Guidelines for National Greenhouse Gas Inventories," Geneva, Switzerland, 2019.

Supplemental

52: For transparency, it would be helpful to see a table with the average GHGI EF, the average EF from this study, and the average EF from all of the input studies. This will help give a better sense of whether it is the approach or the use of a different data source that is driving the difference.

Author: Note that we include tables with average component-level emission factors (Table S2), equipment-level emission factors (Table S3-S4), and average component-level emission factors across the various studies (Table S5) in the supplementary information. We add a note here on L52 to direct the reader to these figures.

SI L57: (see comparison of component-level emission factors in **Table S2**, comparison of equipment-level emission factors in **Tables S3** and **S4**, and a summary of component-level emission factors across the various studies in **Table S5**).

174: If there aren't oil EFs in this study, how was the model run? How was the comparison made? Is most of the gap oil? If so, what evidence is there? For subpart W columns, provide footnotes explaining the difference between leaker methods i and ii. How are the GHGRP splits between east and west taken into account?

Author: The emission factors for our study referenced in Table S2 are undifferentiated and apply to all categories (gas, oil). This has been clarified in the Table caption. We also add footnotes describing the difference between GHGRP leaker methods.

SI L249:

Table S2: Comparison of component-level emission factors (units of kgCH₄/component/day) generated in this study with 1990s EPA emission factors, including API 4589 [16], Star Environmental [29], and the EPA Protocol document [38], and emissions factors applied in Subpart W [34]. Note that emission factors for this study are undifferentiated and apply to all product categories (gas, light oil, heavy oil). Population emission factors (Pop.) are calculated as the average across both leaking and non-leaking components, while “leaker” emission factors are an average of only leaking components. In both this study and the EPA Protocol document, “leaker” emission factors are calculated according to a threshold concentration screening value (see Section 4.2.4).

	API 1993 (doc. 4589) [16]	Star Environmental [29]		EPA Protocol Document [38]			Subpart W [34] ¹			This study (emission factors undifferentiated across all product categories)		
		Pop.	Leaker	Pop.	Ave. >10,000 ppmv	Pegged > 10,000 ppmv	Pop.	Leaker method (i) ²	Leaker method (ii) ³	Pop.	Ave. > 10,000 ppmv	
Gas	Valve	0.044	0.007	0.544	0.074	1.616	1.055	0.010	1.782	1.273	0.047	6.205
	Connector	0.002	0.001	0.073	0.003	0.429	0.462	0.001	0.473	0.291	0.006	3.326
	OEL	0.004	0.020	0.116	0.033	0.907	0.495	0.022	1.018	0.691	0.021	2.356
Light oil	Valve	0.005	-	-	0.037	1.434	-	0.018	1.164	0.800	-	-
	Connector	0.001	-	-	0.003	0.429	-	0.003	0.364	0.218	-	-
	OEL	0.010	-	-	0.021	0.725	-	0.018	0.582	0.400	-	-
Heavy oil	Valve	8.6E-05	-	-	1.9E-04	-	-	1.8E-04	1.164	0.800	-	-
	Connector	4.3E-05	-	-	1.7E-04	-	-	1.1E-04	0.364	0.218	-	-
	OEL	4.3E-04	-	-	3.2E-03	4.9E-01	-	2.2E-03	0.582	0.400	-	-

¹See Tables W-1A for population emissions factors and W-1E for leaker emissions factors in [34]

²Leak detection is conducted using an optical gas imaging instrument as specified in Subpart W Section 98.234(a)(1)

³Leak detection is conducted using Method 21 as specified in Subpart W Section 98.234(a)(2).

185: Were the 2015-specific emissions data used for tanks?

Author: Yes, we now provide a reference to Section 6.3.3 where RY2015 Subpart W data was used to develop tank-basis emission factors for comparison with estimates from this study.

SI L266: Because emission factors for storage tanks are reported by the GHGI on a throughput basis (kg/Mbbl), we perform a rough conversion for comparison with this study's emission factors (see Section 6.3.3 where we describe how reporting year 2015 Subpart W data was used to develop tank-basis emission factors for comparison with estimates from this study).

188: Provide additional information on the approach for assigning EFs/component counts to marginal wells. How many data points are included in each category? Do all of the input studies provide information on production level from the wells measured?

Author: See comment above referencing line 509 (original document location)

195: Why are the completion and workovers emissions so different from GHGI? Shouldn't these be the same or at least more similar since they are using the same data source? Does this indicate an issue with the modeling?

Author: The reason our completion and workover emissions are different from the GHGI is because the GHGI computes combined completion and workover emission factors (e.g., for each category, summed completion and workover emissions are divided by summed completion and workover activity). This is in contrast to our approach, where separate emission factors are computed. This is now clarified in text.

Note that we have made other adjustments to our calculation of completion and workover emission factors. Specifically, we now use reporting year 2016 data from GHGRP. Although this doesn't align with our basis year (2015), 2016 is the first year that oil wells reported. This allows us to calculate emission factors specific to natural gas systems and petroleum systems (before, natural gas system emission factors were also assigned to petroleum systems). We believe this is a more accurate approach.

SI L990: For each category, we sum reported CH₄ emissions and divide by the number of reported completion or workover events. Note that we use GHGRP reporting year 2016 data to develop differentiated oil and gas emission factors. 2016 is the first year oil wells were required to submit data on completions and workovers to the GHGRP [42]. Even though we use the same data source, our emission factors are still different from the GHGI. This is because the EPA develops combined completion and workover emission factors (in contrast to our approach, where the two source categories are separated, [43]). It is not clear from documentation why this approach was chosen.

Examining activity data in the 2020 GHGI, GHGRP subpart W counts are applied for some emission sources (e.g., natural gas system hydraulic fracturing completions) but not others (e.g., natural gas system non-hydraulic fracturing completions). This could be because subpart W reporting is still incomplete for certain categories (presumably, not

all well completions are reported by operators). Therefore, for our estimation of total emissions we use emissions factors calculated using subpart W data and AFs from GHGI (Table S12).

Based on these calculations we find total completion and workover CH₄ emissions to be 46.9 Gg for the natural gas sector and 83.6 Gg for the petroleum sector.

Table S12: 2015 emissions from completions and workovers based on emissions factors derived from GHGRP subpart W data and activity factors derived from the GHGI

	Emissions factor [tonne/event]	Activity factor [events/year]	Total [tonne/year]
Natural gas systems			
HF Completions - Non-REC with Venting	8.103	105	850.8
HF Completions - Non-REC with Flaring	0.559	326	182.6
HF Completions - REC with Venting	6.404	3053	19550.6
HF Completions - REC with Flaring	5.500	1795	9871.7
Non-HF Completions - vented	22.656	602	13631.2
Non-HF Completions - flared	0.192	188	36.2
HF Workovers - Non-REC with Venting	4.339	199	864.6
HF Workovers - Non-REC with Flaring	1.266	66	84.1
HF Workovers - REC with Venting	0.516	1833	945.3
HF Workovers - REC with Flaring	0.812	339	275.0
Non-HF Workovers - vented	0.074	7303	537.4
Non-HF Workovers - flared	0.073	349	25.7
Petroleum systems			
HF Completions: Non-REC with Venting	43.664	1494	65233.5
HF Completions: Non-REC with Flaring	1.266	1517	1920.8
HF Completions: REC with Venting	0.899	3630	3262.2
HF Completions: REC with Flaring	1.634	5494	8979.8
HF Workovers: Non-REC with Flaring	13.721	267	3663.6
HF Workovers: REC with Flaring	0.598	976	583.3

195: What is the combustion emission source? In GHGI, combustion emissions are included in a different section of GHGI (fossil fuel combustion). Is it methane slip?

Author: Combustion here is referring to methane slip. The row title has been changed to methane slip.

217: Since the GHGI was updated to use GHGRP data, this is no longer relevant. The Marchese adjustment was from before the GHGI had data (from GHGRP) to allow for a distinct gathering segment and instead had any gathering emissions included in with onshore production.

Author: The authors thank the reviewer for this clarification of the GHGI activity data. The activity factors have been modified to reflect this. See the edited text referenced in the next comment.

220: The Allen et al. counts of pneumatics per well were from a far smaller sample size compared to GHGRP, which has pneumatic controller counts reported every year from hundreds of thousands of wells. Consider updating the model to use the GHGRP data set.

Author: The authors appreciate the reviewer's clarification of the Allen et al study. We now use exclusively GHGRP data for all activity factors. Our model and supplementary information have been adjusted to reflect this. The text referenced here has been deleted and the paragraph has been updated as follows:

SI L302: Equipment-level activity factors applied in this study are generally very close to those applied in the GHGI. We generated equipment-level activity factors using the same approach as the GHGI (see description of approach using GHGRP data in Section 4.4), so ideally our values should be identical. Updates to the GHGRP Envirofacts system between when the GHGI generated activity factors and when activity factors were generated for this study leads to small differences (generally < 5%, although a larger adjustment of ~35% in Envirofacts appears to have been made to reciprocating compressor activity).

309: Here it is noted that few if any component-level studies break out data by production volume. How/why then is the model breaking the data into groups based on production?

Author: See comment above referencing line 509 in the main text

346: Input data from Canada are included. Note the exception here (leak counts I believe) and how it effects the conclusions--here the authors say that differing regulatory and operating standards could impact results; yet the Canadian data are included for equipment leaks where one of the key discrepancies was observed--what role did the use of Canadian data potentially play?

Author: See a previous comment. Adjustment was made at SI L453.

457: Clarify how oil versus gas and different production level EFs were assigned, considering the limited data availability.

Author: See comment above referencing line 509 in the main text

536: Recommend using the GHGRP data on pneumatic controller counts

Author: We now use exclusively GHGRP data for all activity factors. Our model and supplementary information have been adjusted to reflect this. See the activity factor section below which has been modified to reflect these changes.

SL Section 4.4

In the GHGI, direct equipment counts are not available for every year. As an approximation, the GHGI uses “activity drivers” (AD) such as gas production, number of producing wells, or system throughput. AD are multiplied by a scaling factor (e.g., separators per well, af_k^*) derived from a subsample of the population. For each piece of equipment, we employ well counts as the activity driver, AD . Since the 2018 GHGI, the EPA has calculated activity factors for most equipment using scaling factors based on GHGRP data. Scaling factors based upon reporting year 2015 equipment counts are multiplied by year-specific wellhead counts to calculate year-specific equipment counts [44].

$$af_k = AD \cdot af_k^*$$

Equipment-level activity factors are calculated using the same approach as the GHGI (documented in [7], [44]). Therefore, any differences between activity factors of our study and the GHGI are due to relatively minor differences in well counts. This is probably because Enverus well counts (the source for both our study and the GHGI, see Section 5.1.2) are revised on a regular basis). Briefly, equipment count data from EPA’s Envirofacts was used (reported by operators under Subpart W). Equipment count data for reporting year 2015 was downloaded from several tables:

“EF_W_ATM_STG_TANKS_CALC1OR2”, “EF_W_ATM_STG_TANKS_CALC3”, “EF_W_NGPNEUMATIC_DEV_UNITS”, and “EF_W_EQUIP_LEAKS_ONSHORE”.

Activity scaling factors were calculated by dividing reported equipment counts by reported wells, separately for natural gas systems and petroleum systems (Table S13).

These scaling factors, af_k^* , are applied for each equipment k . Because our algorithm iterates across equipment count, the need for AD in the formula explicitly disappears. For each bin, i , emissions are calculated well-by-well. For a single well, j , equipment-level emissions are calculated by multiplying a randomly drawn emissions factor, $EF_{i,j,k}$ [kg/equipment/day], by its respective activity scaling factor, af_k [# equipment/well]. In the OPGEE CH₄ emissions algorithm, af_k^* is represented as follows:

$$Q_{population} = \sum_{i=1}^{n_{bins}} \left\{ \sum_{j=1}^{n_{wells,i}} \left[\sum_{k=1}^{n_{equip}} EF_{i,j,k} * X \right] \right\}$$

Where,

$$X = \begin{cases} 1 & p \leq af_k^* \\ 0 & p > af_k^* \end{cases}$$

Table S13: Scaling factors, af_k^* , applied

Equipment	Scaling factor (count per well) ¹
Natural gas system	
Heater	0.13
Separator	0.71
Dehydrator	0.03
Meters	0.84
Small Recip compressor	0.08
Chemical Injection Pumps	0.18
Pneumatic controllers	1.87
Tanks	0.41
Petroleum system	
Heater-treater	0.19
Separator	0.37
Header	0.22
Chemical Injection Pumps	0.10
Pneumatic controllers	1.11
Tanks	0.82

¹Scaling factors are based on counts of equipment divided by counts of wellheads in several tables reported on the EPA Envirofacts website, reported under Subpart W for reporting year 2015. For tanks, data is from “EF_W_ATM_STG_TANKS_CALC1OR2” and “EF_W_ATM_STG_TANKS_CALC3”. For pneumatic controllers, data is from “EF_W_NGPNEUMATIC_DEV_UNITS”. Chemical injection pumps are reported an EPA memo [44]. All remaining scaling factors are based on equipment counts reported in “EF_W_EQUIP_LEAKS_ONSHORE”.

634: Does this approach potentially underestimate the EF for marginal wells? E.g. leak detection may be less prevalent at marginal wells so it may not necessarily be the case that they have much lower emissions than nonmarginal wells.

Author: See comment above referencing line 509 in the main text

636: How are the means higher than the two subcategories?

Author: “Mean” was not the correct word here. These are better described as “total” emission factors. This section has been rewritten to better describe how the emission factors are calculated.

SL Section 4.3.1

The “stochastic failure” approach is used to estimate emissions from: all equipment leakage emissions and abnormal process vents from tanks. In the stochastic failure approach, a sample of component-level measurements are iteratively re-sampled

(bootstrapped) to generate a distribution of equipment-level emission factors. Here, we will briefly explain the development of equipment level emission factor distributions.

The general stochastic failure equation is as follows. For a single piece of equipment:

$$EF = \sum_{k=1}^{n_c} \sum_{l=1}^{LQ_k} CF_k$$

Based upon the number of leaking components, LQ_k , a corresponding number of leaks of the correct component type are randomly drawn from the database of CF (with replacement). The equipment level emission factor is then calculated by summing across all components.

Recall that our dataset has been split at a threshold of 10,000 ppmv (describing leaks that were missed by optical gas imaging but caught with Method 21 below the threshold, and leaks that were caught with optical gas imaging above the threshold, see Section 4.2.4). Therefore, we can rewrite the previous equation as follows:

$$EF = \sum_{k=1}^{n_c} \left\{ \sum_{l=1}^{LQ_{k'}} (CF_k)_{lo} + \sum_{l=LQ_{k'}}^{LQ_k} (CF_k)_{hi} \right\}$$

We have defined this split to (to the best of our ability) describe two distinct and mutually exclusive parts of the leaker distribution. Here the subscript 'lo' refers to the portion of the dataset tagged at <10,000 ppmv, and the subscript 'hi' refers to the portion of the dataset that were tagged above that threshold.

Next, we must define LQ' , which describes the number of leakers that are drawn in the 'lo' set versus the 'hi' set. First, we will define the fraction leaking value, FL , as follows:

$$FL = \frac{LQ}{CQ}$$

Where CQ is the total components on a particular piece of equipment.

Earlier, when we derived fraction leaking values for the 'lo' and 'hi' concentration ranges (Section 4.2.4) based upon our source studies, we did this by splitting up the datasets according to the Method 21 and optical gas imaging techniques (See **Table S10**).

Therefore,

$$(FL_{total})_k = \frac{(LQ_{total})_k}{CQ_k} = \frac{(LQ_{lo})_k}{CQ_k} + \frac{(LQ_{hi})_k}{CQ_k} = FL_{lo} + FL_{hi}$$

So, by the same logic:

$$LQ_{lo} = CQ \times FL_{lo}$$

$$LQ_{hi} = CQ \times FL_{hi}$$

Resetting the indices in our primary equation:

$$EF = \sum_{k=1}^{n_c} \left\{ \sum_{l=1}^{CQ \times FL_{lo}} (CF_k)_{lo} + \sum_{l=1}^{CQ \times FL_{hi}} (CF_k)_{hi} \right\}$$

Therefore, the complete distribution can be defined as the superposition of the two halves, defined by separate sets of measured leaks and separate fraction leaking values

$$EF = \sum_{k=1}^{n_c} \sum_{l=1}^{CQ \times FL_{lo}} (CF_k)_{lo} + \sum_{k=1}^{n_c} \sum_{l=1}^{CQ \times FL_{hi}} (CF_k)_{hi}$$

$$EF = EF_{lo} + EF_{hi}$$

Note that for this manuscript, this process is adjusted such that the number of components leaking per piece of equipment (LQ) is tallied according to a random draw. This process is repeated 10,000 times ($n_{\text{trials}} = 10,000$) to develop emission factor distributions. In iterative loops, the algorithm sweeps across all equipment (n_e), all components (n_c), and all components per equipment (CQ).

$$LQ_{i,j,k} = \sum_{l=1}^{CQ_{i,j,k}} X_{i,j,k,l} \quad \forall i \in \{1, \dots, n_{\text{trials}}\}, j \in \{1, \dots, n_e\}, k \in \{1, \dots, n_c\}$$

Here LQ is a 3-dimensional matrix, with dimensions $10,000 \times n_e \times n_c$. During the iteration across each component class, X is a random binary variable set equal to 1 if a uniformly distributed random number (p) on $[0,1]$ is drawn that is greater than FL for that component type. Put differently, for each leak draw l , if the random number drawn is less than the probability of leakage then LQ_{ijk} is incremented by 1.

$$X_{i,j,k,l} = \begin{cases} 1 & p \leq FL \\ 0 & p > FL \end{cases}$$

The resulting matrix LQ_{ijk} is the number of leaks for each component type k in each equipment type j , for each realization i (of which there are 10,000).

Next, the algorithm assigns leakage volumes to these leaks. Based upon the number of leaks in LQ, a corresponding number of leaks of the correct component type are randomly drawn from the database of CF (with replacement). This process is repeated 10,000 times producing a second 3-dimensional matrix with dimensions $10,000 \times n_e \times n_c$

$$EF_{i,j,k} = \sum_{l=1}^{LQ_{i,j,k}} CF \quad \forall i \in \{1, \dots, n_{\text{trials}}\}, j \in \{1, \dots, n_e\}, k \in \{1, \dots, n_c\}$$

The matrix EF is then reduced to 2 dimensions by summing across components, or summing across index k . This results in a new matrix $EF_{i,j}$ with values of emissions for 10000 trials (index i) and n_e equipment types (index j). The Matlab tool "prctile" is used to produce probability distributions of emissions per piece of equipment using this matrix.

After superposing the separate equipment-level distributions for < 10,000 ppmv measurements and ≥ 10,000 ppmv measurements, the resulting distributions are embedded in OPGEE’s CH₄ emissions calculator as a 0 - 100 percentile table for each equipment type *j* in 1...*n_e*. Because we have separate component counts for gas and oil systems, we also have separate equipment level emission factor distributions.

It should be noted that the emissions-factor distributions in **Table S12** (*a priori* distributions) are different from the emissions factor distributions presented earlier in **Table S3** and **Table S4** (*post-hoc* distributions). The distributions presented below are eventually applied to OPGEE’s CH₄ emissions algorithm (Section 4.1), where OPGEE iteratively assigns *a priori* emission factors. However, due to the conservation of mass checks, some emission factor draws are not assigned. *Post hoc* emissions-factors are calculated according to the actual emissions randomly assigned by OPGEE’s iterative bootstrapping algorithm. Based on the OPGEE conservation of mass check, *post-hoc* emissions factors are lower than the average *a priori* emissions-factors. This also results in marginal wells emission factors much lower than non-marginal wells emission factors.

Table S12: Averages of *a priori* equipment level emissions factor distributions [kg/day].
“Total” emission factor distributions are the superposition of the “500-10,000 ppmv” and “>10,000 ppmv” distributions.

	Natural gas systems			Petroleum systems		
	Total	500-10,000 ppmv	> 10,000 ppmv	Total	500-10,000 ppmv	> 10,000 ppmv
Well heads	4.6	1.3	3.3	3.2	1.0	2.2
Header	4.1	1.2	2.9	9.8	2.9	6.9
Heater	3.8	1.0	2.8	2.5	0.8	1.7
Separator	4.6	1.1	3.4	3.7	1.0	2.7
Meter	3.6	0.8	2.9	2.5	0.6	1.9
Tanks - Leaks	2.9	0.9	1.9	1.7	0.5	1.2
Tanks - Vents	12.3	1.2	11.1	12.4	1.2	11.2
Compressor - Recip	6.7	2.0	4.7	2.4	0.7	1.7
Dehydrator	4.8	1.2	3.6	2.0	0.6	1.4
Chemical Injection Pump	7.1	0.0	7.1	7.0	0.0	7.0
Pneumatic Controller	3.2	0.1	3.2	3.2	0.1	3.2

670: NSPS does not require controls at all tanks. It requires controls at new or modified tanks over a certain threshold.

Author: Thank you for this insight. The wording has been clarified.

SL 831: For tanks equipped with controls, flashing emissions are destroyed or captured. US EPA NSPS subpart OOOO requires all new or modified tanks (as of April 2013) to install control devices such as flares or vapor recovery units designed to reduce volatile organic compound emissions by at least 95% [39].

693: Note that the # of wells onsite is lower here than for many site-level studies. What does this mean for comparability of the results with other studies?

Author: This is true. Omara et al. acknowledge in their synthesis study that, compared to the overall US average (1.64 wells/site), site-level studies tend to sample higher production sites. This is now acknowledged in our description of the Omara and Alvarez papers in the supplementary information Section 2 (see full section above, relevant section copied below).

SL 179: Note that the distribution of sampled sites in Omara et al. are generally higher production compared to the overall population of US sites (Fig S21 in Omara). Although this is accounted for in the model, low producing sites may not be characterized as well as high producing sites.

724: Provide rationale for why Allen et al was used for liquids unloading as opposed to GHGRP.

Author: For this paper, our goal was to construct a dataset consists entirely of measured data. GHGRP data for liquids unloadings are modelled. We have made this pointer clearer in the supplementary information section "Summary of criteria for inclusion in our analysis".

SL 436: We reviewed the literature for field studies with measurement and quantification of CH₄ emissions data at the component scale. All studies with emission measurement performed at the wellsite/facility or producing field/region scale were removed from consideration. To the extent possible, we only include studies with component-level direct measurements (e.g., using a High Flow Sampler) and avoid studies with simulated emissions data (e.g., based on engineering equations).

791: Why are the values for completions and workovers so different from GHGI if they are using the same dataset?

Author: See previous comment for SI L195.

793: The emission factors do not seem to match those used in the GHGI.

Author: See previous comment for SI L195.

810: See previous comment—Marchese adjustment no longer needed.

Author: The authors thank the reviewer for this clarification of the GHGI activity data. The activity factors have been modified to reflect this (See quoted text at previous comment citing SI L536).

821: The equipment should all be allocated to production.

Author: The authors thank the reviewer for this clarification of the GHGI activity data. The activity factors have been modified to reflect this (See quoted text at previous comment citing SI L536).

821: Recommend use of GHGRP for pneumatics per well for reasons noted above. In addition, this may be a misinterpretation of the Allen et al counts. A previous EPA assessment estimated 1.5 venting pneumatics per well based on Allen et al.: “The Allen et al. study observed 2.7 controllers per well surveyed. Only sites with venting pneumatic controllers were visited for the study. However, the study notes that some sites use non-venting and/or non-pneumatic controllers, and that therefore 2.7 controllers per well represents an upper bound. To develop rough estimates of national emissions, in one scenario Allen et al. assumed that 75% of all controllers were pneumatic-style controllers, of which 75% would be actively venting methane. This assumption results in an estimated average of 1.5 venting pneumatic controllers per well.” <https://www.epa.gov/sites/production/files/2015-12/documents/ng-petro-inv-improvement-pneumatic-controllers-4-10-2015.pdf>

Author: The authors appreciate the reviewer’s clarification of the Allen et al study. We now use exclusively GHGRP data for all activity factors. Our model and supplementary information have been adjusted to reflect this (See quoted text at previous comment referencing pneumatic controllers, citing SI L536).

838: Composition data is also available from GHGRP. The GHGI predominantly (though indirectly) uses CH₄ concentration data from GHGRP (by using mass of CH₄ reported to GHGRP for many sources).

Author: This is a good suggestion. We have updated our values for CH₄ mole fraction based on data reported to GHGRP.

SI L1049: The CH₄ content reported in the EPA GHGI ([40], Table 3.6-3) is itself based upon the Gas Technology Institute Unconventional Natural Gas and Gas Composition Databases [45]. The 2015 average CH₄ content, weighted by regional production, reported in the GHGI is 82.2% (volume basis).

However, as GHGRP data is becoming more widely used for GHGI source categories, this reported gas composition gets applied indirectly (operators are required to report CH₄ content under Section 98.236(aa)(1)(ii)). The CH₄ mole fraction used in OPGEE is based on average mole fractions from reporting year 2015 Table “EF_W_FACILITY_OVERVIEW”. Using the same approach as the EPA in partitioning facilities (summarized in Section 6.3.1) we compute average mole fractions for natural gas and petroleum systems, 83.2% and 68.3%, respectively. These values are applied in OPGEE.

860: Specify which EPA program uses these 4 categories. The GHGI does not.

Author: This additional separation (from 2 categories to 4 categories) was not done for consistency with the EPA. We made additional adjustments for reasons specific to our model. This is now clarified.

SI L1095: First, the dataset was split into four groups applying the same GOR cutoff as the EPA [41]: (i) Gas only wells, (ii) gas wells with oil production (GOR > 100 Mscf/bbl), (iii) oil wells with associated gas (GOR < 100 Mscf/bbl), (iv) and lastly oil only wells (no gas production reported). Note that the EPA does not distinguish oil wells and gas wells from associated wells (i.e., oil wells with gas production and gas wells with oil production), but this separation was conducted for our analysis. Dry gas wells are

analyzed in a separate category from gas wells with oil production because unique treatment is required in OPGEE (OPGEE requires a nominal amount of liquids production to simulate dry gas wells). Oil wells are analyzed separately from oil wells with associated gas for the purposes of an additional calculation step (described in Section 5.1.4, we assume a small amount of associated gas is produced, even if no marketed gas is reported).

965: Is there variation in the wells per site in each of these categories? Does that impact the comparisons with site-level data?

Author: There is variation in wells per site across product class and productivity tranche. Because our Enverus dataset contains information on the well-to-site clustering performed by Alvarez in analyzing site-level emissions we are able to capture this variability in our model. We have added this clarification.

SI L1308: We iterate across rows (well, production, emissions combinations) in the OPGEE outputs and randomly assign a bin-specific “well-count”. Because OPGEE emissions outputs and the Enverus well count data have been binned in the same manner, any variation in wells per site across product class and productivity tranche is captured in our model.

1072: Were all petroleum systems CH₄ emissions included in the numerator or just a fraction that was attributed to gas production versus oil production? Another option is to do a BOE comparison.

Author: This loss rate includes all CH₄ emissions from the petroleum system production segment (i.e., no allocation was performed). We have added a note to the figure caption.

SI L1511: Figure S15: Historic time series of methane emissions according to the GHGI [40]. Loss rate calculated as O&NG production CH₄ emissions divided by gross CH₄ withdrawals (using EIA data and a CH₄ fraction of 82.2%, [45], [46]). Note that the loss rate includes all CH₄ emissions from the petroleum system extraction and production segments (i.e., no allocation was performed between natural gas and petroleum systems).

1083: This is true only for equipment leaks. For sources calculated with more recent data (basically all here except equipment leaks) those are considered net emissions.

Author: We have clarified this in the text.

SI L1523: Given difficulties in obtaining reductions by source category, emissions reductions are typically reported by supply chain segment. However, based on incorporation of new data (described below), these reductions are largely concentrated in the equipment leaks category. Any comparisons made in this study are with potential emissions before mitigation.

1090: The first emissions year of GHGRP reporting for natural gas and petroleum was 2011 (reported in 2012), not 2009. The GHGI began incorporating those data in 2014.

Author: Thank you for the correction. We have edited the text accordingly.

SI L1532: Since 2010 (reported in 2011), through the GHGRP the EPA has mandated that industrial facilities emitting > 25,000 tonne/y in CO₂eq. terms must report GHG data. Data is available for petroleum and natural gas systems beginning with reporting year 2011 (reported in 2012).

1102: Note that the current GHGI (using GHGRP) values for year 2012 are still consistent with the Allen et al values measured in 2012. As more recent data from GHGRP continue to become available, they (for the most part) show a decreasing trend from 2012.

Author: We agree that updating inventory values for liquids unloadings with GHGRP data had a minor impact on emissions for 2012. We have updated the text to more accurately describe an “update” instead of a “revision downward”. We also note that the 2012 estimate is still consistent with Allen et al. [3].

SI L1552: CH₄ emissions estimates for liquids unloadings were updated based on GHGRP data [7]. Prior to this adjustment, Allen et al. [3] had noted that GHGI estimates were already in close agreement with field observations. In the most recent GHGI, 2012 liquids unloadings are still in close agreement with Allen et al. Emissions from liquids unloadings have decreased over time from 263 Gg to 177 Gg in 2018 [40].

1132: Please recheck the references. The GRI study was commissioned by GRI and EPA, and conducted (in part at least) by Radian. Unclear about STAR involvement.

Author: This statement was intended to refer to API publication 4598 and the 1995 Eastern Gas Wells study, which were both conducted by STAR. However, the relationship between the GRI study and the STAR reports is much clearer in the following sentences. Thus, the referenced statement has been edited to minimize confusion.

SI L1602: The structure of the GHGI is rooted in a suite of studies from the 1990s ([16], [29], [47]). Some emissions sources have been updated using data from the GHGRP (see previous section), however many sources are still based on this original data. The GHGI cites a 1996 report by the Gas Research Institute ([30], henceforth referred to as the “GRI report”) for natural gas systems and a 1996 calculation workbook by the American Petroleum Institute ([31], henceforth referred to as “API 4638”) for petroleum systems (see the first branch connecting the GHGI to these studies in **Figure S16**).

1145: It might be helpful to label here which references are used in oil versus gas.

Author: This is a good idea. We have added labels to the Figure.

Figure S16: Flow chart representing relationships between various cited studies in the GHGI [40]. The GRI study [30] and API [31] are cited by the GHGI in the development of component and equipment-level emission factors. These two studies cite Star Environmental [29] and API [16] for primary, component-level emissions data.

1173: Differences in the regional breakout of well locations and regional CH₄ content is likely what drives the differences.

Author: The authors agree this is likely to be the source of variation year over year. We have incorporated this suggestion.

SI L167: The GHGI reports adjustments to emission factors year over year. It is generally not clear in the GHGI documentation why these adjustments were made, but a likely explanation is adjustments to regional shares of well counts and gas composition.

1215: Are there potentially issues with the correlation equations? Are there recommendations for improvements?

Author: We note that we do not examine the statistical aspects of the correlation equations in detail, but add a brief comment at this point in the text.

SI L1750: We don't examine in detail the statistical properties of the correlation equation approach, but note that direct quantitative measurements (for example, using a high flow sampler) should be preferred to the correlation equation approach. Caution should especially be used given the treatment of large emitters using "pegged factors", given the likely importance of the top 5% (super-emitters)

1362: This is a key caveat. There is a chance the difference is due to the studies representing different regions, types of systems, etc. This should be clearer in the main body of the paper.

Author: See discussion at the beginning of the review. We have added text to acknowledge limitations to the representativeness of our data (manuscript L119, L164, and L467)

1394: Clarify which of the GHGI data sets are also included in this study.

Author: This is an important point to clarify. We have added text to both the SI text and figure caption.

SI L1868: We also compare our dataset separately with data underlying the GRI report's (i) Western and petroleum systems and (ii) Eastern systems. For Eastern systems, low emitters (< 10,000 ppmv) are based upon the Star Environmental [29] dataset combined with EPA Protocol Document correlation equations [38]. Higher emitters (> 10,000 ppmv) are based solely upon High Flow sampler measurements made for the Star Environmental study. For Western and petroleum systems, low emitters are based upon API 4589 [16] combined with the EPA Protocol Document correlation equations. High emitters are based upon pegged source factors developed in the EPA Protocol Document [38]. For this section, rather than simply comparing with the pegged source factor, we present the full distribution of emissions measurements used to generate the pegged source factor. Recall that pegged source factors were developed based on quantified emissions measurements (High Flow sampler) from refinery, marketing terminal, and oil and gas production facilities (measurements digitized from the EPA Protocol document Appendix C Attachment 2, [38]). It should be noted that quantified emissions measurements from API 4589 (production segment only) are also contained in our study.

SI L1897: Figure S19: Component-level emissions aggregated across all production-segment sources comparing the Western US and petroleum data set (API 4589 and EPA protocol document) [16], [38](a and b) and the Eastern US data set (Star Environmental) [29] (c and d) with our dataset. Western and Eastern datasets are disaggregated into component-level emissions underlying pegged factors for comparison with our dataset $\geq 10,000$ ppmv (a and c) and emissions generated from correlation equations (corr.) are compared with our dataset < 10,000 ppmv (b and d). Rather than comparing with the pegged source factors (a single number), in panel (a) we compare our dataset with the underlying emissions measurements used in developing the pegged source factors (digitized from EPA Protocol document Appendix C Attachment 2, [38]). Note that quantified measurements from API 4589 [16] are present in both our dataset and the EPA Protocol document pegged source factor dataset. Horizontal bars span the 5th and 95th percentile. Note that the histogram scale for this study's dataset is on the left-hand side of the plots, and the scale for the GHGI datasets is on the right hand side of the plots. Note the log scale of emissions with the implication that discrepancies span 1-3 orders of magnitude.

1434: These are important points and should be covered in the main body

Author: We have added very similar text to an appropriate spot in the main manuscript.

Manuscript L318: We can only speculate as to why this difference exists, but possibilities include sampling bias in the original collection process or fundamental differences in the populations sampled in the EPA’s basis datasets versus those in this study (for example, most O&NG is now produced from unconventional shale formations whereas it wasn’t during the time of the original GRI study).

1456: Is this all of the equipment at the well or just wellheads?

Author: This is referring to the wellhead (equipment type) only. We have adjusted the wording on all figures that reference wellheads.

1465: Is the GHGI value for # leakers missing here?

Author: Both our approach and the GHGI approach differentiate between leakers >10,000 ppmv (large leaks) and leakers 500-10,000 ppmv (small leaks). In order to simplify the decomposition plots, we limit component-level emission factors and fraction of components emitting to large leaks only. Because API 4589 only screened small leaks for heavy oil systems, the fraction of components emitting in Figure S24 is zero (although the equipment-level emission factor will be non-zero, because small leaks were screened).

This has been clarified in the added section.

SI L1956: Recall that, for this study, equipment-level emission factors are the superposition of “large emitters” (>10,000 ppmv) and “small emitters” (500-10,000 ppmv), each defined by a separate set of measured leaks and fraction of components emitting (see discussion in Section 4.3.1). Similarly, for EPA GHGI construction of equipment-level emission factors, “large emitters” are characterized by pegged source factors and “small emitters” are characterized by correlation equations (see discussion in Section 6.2.2). For illustrative purposes, and because “large emitters” (>10,000 ppmv) constitute the majority of total emissions, our decomposition plots do not include component-level emission factors and fraction of components emitting for “small emitters”. Thus, total equipment level emission factors will not calculate exactly as the product of the constituent parts. For example, for heavy oil wellheads in **Figure S24**, the fraction of components emitting is 0. This is because in the API 4589 [16] survey, no components were found emitting at heavy oil wellheads at >10,000 ppmv. However, the equipment level emission factor is still non-zero because components were found emitting at 500-10,000 ppmv.

1672: The malfunctioning dump valves are reported for controlled and uncontrolled large tanks and are not reported for small tanks.

Author: This statement has been removed. Table S29, where malfunctioning separator dump valves were mistakenly attributed to small tanks, has been adjusted.

1680: What is it that’s inherent to the simulation approach that results in an underestimate?

Author: What this statement is referring to is the observation that there are some simulated volumes over an order of magnitude lower compared to the lowest measured volume in our dataset. This is because measured emission volumes are limited by the lower detection limit of

the measurement device, whereas simulated volumes are not. This point is now clarified in the text.

SI 2262: Note that that there are some simulated volumes over an order of magnitude lower compared to the lowest measured volume in our dataset. This is because measured emission volumes are limited by the lower detection limit of the measurement device, whereas simulated volumes are not.

1686: Could another approach to address the gap be to include an improved EF for unintentional events? If that is where most of the difference is, this could be a more straightforward approach.

Author: This is a relatively simple and straightforward approach which we have added to the discussion section of the main text.

Manuscript L501: Because all emissions data and activity factors (with some exceptions, noted in methods) are US-based, emission factors from this study (summarized in **Tables S2-S4**) could be implemented in US inventories. Emission factors for equipment leaks could be implemented relatively easily by updating existing sources categories. Implementing emission factors from storage tanks based on this study would require modifications to source categorization, for example, through the addition of a new factor to take into account failed controls like open thief hatches.

1933: There isn't much discussion in the paper about the split between eastern and western regions for equipment leaks in the GHGI, established in the 1992 GRI study. Is it possible that shifts in production and practices across the U.S. compared to 1992 also contribute to the gap?

Author: See discussion at the beginning of the review. We have added text to acknowledge this (at manuscript L346).

Review response bibliography

- [1] T. L. Vaughn *et al.*, "Temporal variability largely explains top-down/bottom-up difference in methane emission estimates from a natural gas production region," *Proc. Natl. Acad. Sci. U. S. A.*, 2018, doi: 10.1073/pnas.1805687115.
- [2] R. A. Alvarez *et al.*, "Assessment of methane emissions from the US oil and gas supply chain," *Science (80-.)*, p. eaar7204, 2018.
- [3] D. T. Allen *et al.*, "Methane emissions from process equipment at natural gas production sites in the United States: Liquid unloadings," *Environ. Sci. Technol.*, 2015, doi: 10.1021/es504016r.
- [4] D. T. Allen *et al.*, "Methane emissions from process equipment at natural gas production sites in the United States: Pneumatic controllers," *Environ. Sci. Technol.*, 2015, doi: 10.1021/es5040156.
- [5] D. T. Allen *et al.*, "Measurements of methane emissions at natural gas production sites in the United States," *Proc. Natl. Acad. Sci. U. S. A.*, 2013, doi: 10.1073/pnas.1304880110.
- [6] (EPA) Environmental Protection Agency, "Revisions to Natural Gas and Petroleum Production Emissions," 2016. [Online]. Available: https://www.epa.gov/sites/production/files/2016-08/documents/final_revision_to_production_segment_emissions_2016-04-14.pdf.
- [7] (EPA) Environmental Protection Agency, "Revisions to Natural Gas and Petroleum Systems Production Emissions," 2017. [Online]. Available: https://www.epa.gov/sites/production/files/2017-04/documents/2017_ng-petro_production.pdf.
- [8] M. Omara *et al.*, "Methane Emissions from Natural Gas Production Sites in the United States: Data Synthesis and National Estimate," *Environ. Sci. Technol.*, 2018, doi: 10.1021/acs.est.8b03535.
- [9] M. Omara *et al.*, "Methane Emissions from Natural Gas Production Sites in the United States: Data Synthesis and National Estimate," *Environ. Sci. Technol.*, vol. 52, no. 21, pp. 12915–12925, 2018.
- [10] D. Zavala-Araiza *et al.*, "Toward a Functional Definition of Methane Super-Emitters: Application to Natural Gas Production Sites," *Environ. Sci. Technol.*, 2015, doi: 10.1021/acs.est.5b00133.
- [11] D. R. Lyon, R. A. Alvarez, D. Zavala-Araiza, A. R. Brandt, R. B. Jackson, and S. P. Hamburg, "Aerial Surveys of Elevated Hydrocarbon Emissions from Oil and Gas Production Sites," *Environ. Sci. Technol.*, 2016, doi: 10.1021/acs.est.6b00705.
- [12] (ERG) Eastern Research Group, "City of Fort Worth Natural Gas Air Quality Study," Morrisville, NC, 2011.
- [13] A. R. Brandt, G. A. Heath, and D. Cooley, "Methane Leaks from Natural Gas Systems Follow Extreme Distributions," *Environ. Sci. Technol.*, 2016, doi: 10.1021/acs.est.6b04303.
- [14] D. Zavala-Araiza *et al.*, "Methane emissions from oil and gas production sites in Alberta, Canada," *Elementa*, 2018, doi: 10.1525/elementa.284.
- [15] A. P. Pacsi *et al.*, "Equipment leak detection and quantification at 67 oil and gas sites in the Western United States," *Elementa*, 2019, doi: 10.1525/elementa.368.

- [16] Star Environmental, "Fugitive hydrocarbon emissions from oil and gas production operations. API Publication 4589," 1993.
- [17] Clearstone Engineering Ltd., "Update of Equipment, Component and Fugitive Emission Factors for Alberta Upstream Oil and Gas," Calgary, AB, 2018.
- [18] H. L. Brantley, E. D. Thoma, W. C. Squier, B. B. Guven, and D. Lyon, "Assessment of methane emissions from oil and gas production pads using mobile measurements," *Environ. Sci. Technol.*, 2014, doi: 10.1021/es503070q.
- [19] A. M. Robertson *et al.*, "Variation in Methane Emission Rates from Well Pads in Four Oil and Gas Basins with Contrasting Production Volumes and Compositions," *Environ. Sci. Technol.*, 2017, doi: 10.1021/acs.est.7b00571.
- [20] J. D. Goetz *et al.*, "Atmospheric emission characterization of marcellus shale natural gas development sites," *Environ. Sci. Technol.*, 2015, doi: 10.1021/acs.est.5b00452.
- [21] M. Omara, M. R. Sullivan, X. Li, R. Subramian, A. L. Robinson, and A. A. Presto, "Methane Emissions from Conventional and Unconventional Natural Gas Production Sites in the Marcellus Shale Basin," *Environ. Sci. Technol.*, 2016, doi: 10.1021/acs.est.5b05503.
- [22] T. I. Yacovitch *et al.*, "Mobile Laboratory Observations of Methane Emissions in the Barnett Shale Region," *Environ. Sci. Technol.*, 2015, doi: 10.1021/es506352j.
- [23] X. Lan, R. Talbot, P. Laine, and A. Torres, "Characterizing Fugitive Methane Emissions in the Barnett Shale Area Using a Mobile Laboratory," *Environ. Sci. Technol.*, 2015, doi: 10.1021/es5063055.
- [24] C. W. Rella, T. R. Tsai, C. G. Botkin, E. R. Crosson, and D. Steele, "Measuring emissions from oil and natural gas well pads using the mobile flux plane technique," *Environ. Sci. Technol.*, 2015, doi: 10.1021/acs.est.5b00099.
- [25] J. Penman, M. Gytarsky, T. Hiraishi, W. Irving, and T. Krug, *2006 IPCC - Guidelines for National Greenhouse Gas Inventories*. 2006.
- [26] E. Buendia *et al.*, "Volume 2, Chapter 4: Fugitive Emissions - 2019 Refinement to the 2006 IPCC Guidelines for National Greenhouse Gas Inventories," Geneva, Switzerland, 2019.
- [27] S. Day, M. Dell'Amico, R. Fry, and H. Tousi, "Field Measurements of Fugitive Emissions from Equipment and Well Casings in Australian Coal Seam Gas Production Facilities," *CSIRO, Aust.*, 2014.
- [28] (ERG) Eastern Research Group, "City of Fort Worth Natural Gas Air Quality Study: Revised Final Point Source Test Plan," 2010.
- [29] Star Environmental, "Fugitive Hydrocarbon Emissions: Eastern Gas Wells," 1995.
- [30] M. R. Hummel, K.E., Campbell, L.M. and Harrison, "Methane Emissions from the Natural Gas Industry. Volume 8. Equipment Leaks," 1996.
- [31] Star Environmental, "Calculation Workbook for Oil and Gas Production Equipment Fugitive Emissions. API Publication 4638," 1996.
- [32] (API) American Petroleum Institute, "PRODUCTION TANK EMISSIONS MODEL - A PROGRAM FOR

- ESTIMATING EMISSIONS FROM HYDROCARBON PRODUCTION TANKS - E&P TANK VERSION 2.0," 2000.
- [33] aspentech, "HYSYS 2004 Simulation basis," 2004.
- [34] Code of Federal Regulations, *Title 40 Part 98 Subpart W, Petroleum and Natural Gas Systems*. 2010.
- [35] B. Gidney and S. Pena, "Upstream Oil and Gas Storage Tank Project Flash Emissions Models Evaluation," 2009.
- [36] A. P. Ravikumar *et al.*, "Repeated leak detection and repair surveys reduce methane emissions over scale of years," *Environ. Res. Lett.*, 2020, doi: 10.1088/1748-9326/ab6ae1.
- [37] A. Hendler, J. Nunn, J. Lundeen, and R. McKaskle, "VOC emissions from oil and condensate storage tanks," 2009.
- [38] (EPA) Environmental Protection Agency, "Protocol for Equipment Leak Emission Estimates. Report No. EPA-453/R-95-017," 1995.
- [39] Code of Federal Regulations, *Title 40 Part 60 Subpart OOOOa, Standards of Performance for Crude Oil and Natural Gas Facilities for which Construction, Modification or Reconstruction Commenced After September 18, 2015*. .
- [40] (EPA) Environmental Protection Agency, "Inventory of U.S. Greenhouse Gas Emissions and Sinks: 1990 - 2018," 2020.
- [41] (EPA) Environmental Protection Agency, "Revision to Well Counts Data," 2015.
<https://www.epa.gov/sites/production/files/2015-12/documents/revision-data-source-well-counts-4-10-2015.pdf>.
- [42] (EPA) Environmental Protection Agency, "Inventory of U.S. Greenhouse Gas Emissions and Sinks 1990-2017: Other Updates Considered for 2019 and Future GHGs," 2019.
https://www.epa.gov/sites/production/files/2019-04/documents/2019_ghgi_updates_-_other_updates_2019-04-10.pdf.
- [43] (EPA) Environmental Protection Agency, "Inventory of U.S. Greenhouse Gas Emissions and Sinks 1990-2016: Revisions to Create Year-Specific Emissions and Activity Factors," 2018.
https://www.epa.gov/sites/production/files/2018-04/documents/ghgemissions_year_specific_2018.pdf.
- [44] (EPA) Environmental Protection Agency, "Additional Revisions Considered for 2018 and Future GHGs," 2018. [Online]. Available: www.epa.gov/sites/production/files/2018-04/documents/ghgemissions_additional_revisions_2018.pdf.
- [45] Gas Technology Institute, "Gas Resource Database: Unconventional Natural Gas and Gas Composition Databases," 2001.
- [46] (EIA) Energy Information Administration, "Natural Gas Gross Withdrawals and Production," 2020.
https://www.eia.gov/dnav/ng/ng_prod_sum_dc_nus_mmcfa.htm (accessed Jul. 30, 2020).
- [47] Star Environmental, "Emission factors for oil and gas production operations. API Publication 4615," 1995.

REVIEWERS' COMMENTS<

Reviewer #2 (Remarks to the Author):

The revisions have greatly improved the draft. A few further comments:

Main text:

Line 83: "many of" the underlying data sets

Line 186: most measurement campaigns seem to focus on offsite measurements. You could specify the types of measurement campaigns that would help with the issues identified here (tanks and leaks)

Line 242: one thing to consider though is that while the major equipment activity data are very recent, component count data is largely still from the 1990's (both in GHGI and here)

Line 321: A large fraction of the equipment leaks data used in this study (if I'm correctly understanding, per table 1, 77% of the equipment leak components screened and also a large % of the measured leaks) are the same as used in the GHGI. Maybe there's a point to be made on the value of (and perhaps large impact of?) adding to the data set (so that it now includes HF wells, etc) in this study.

Line 518: Default EFs are "in some cases" based upon... The IPCC GL (both 2019 and 2006) use U.S. but also data from Australia, Canada, Norway, etc.

SI:

Line 178: Recheck # of natural gas well sites used as input in calculating national emissions from Omara study (listed here as 498,000). It's probably lower. The total number of natural gas wells in the U.S. is around 500K, so if 498,000 well sites were assumed in the Omara study, that would mean around 1 well per site, when the average in the U.S. is around 2 wells per site (and the average in the Omara measured population is much higher).

Line 280: Is the study EF for completions averaging out emissions over the course of a year so is a per day EF? GHGI EF is per event (so multiple days) so may want to be clearer on the comparison,

Line 481: Clarify which are leaker EFs versus population EFs

Line 657: Could also note info from Zimmerle et al 2019 on gathering and boosting (impact of measuring intermittent PCs for over 72 hours versus 15 mins)

Line 833: NSPS applies only to new or modified with a potential to emit of 6tpy VOC (so not all tanks)

Line 1221: Flare stacks is only one category of flaring in GHGRP. I think this is described elsewhere in the SI but could also be made clear here.

Line 1233: This isn't consistent with GHGRP data which does show flare stacks at gas wells

Line 1241: Recheck—I believe the flare stack category only includes the flare stacks not reported elsewhere (e.g. tanks, completions)

Below we address each comment point-by-point in indented text, indicating any corresponding changes we made to the manuscript in double-indented text beginning with the term “Manuscript:”, “SI:”, or “Abstract:” as well as the specific location within the document of the update. Line numbers are in reference to attached PDF documents with track-changes on. We have re-numbered all references within this response document, including in excerpts from the text, to use a consistent scheme within the document.

Any changes to the manuscript and supplementary information not referenced in this document were made to conform to the requested formatting guidelines.

Reviewer #2 (Remarks to the Author):

The revisions have greatly improved the draft. A few further comments:

Main text:

Line 83: “many of” the underlying data sets

Author: The text has been edited.

Manuscript L84: Previous studies have noted that many of the underlying data sources of the GHGI were published in the 1990s and may be outdated [1]–[3].

Line 186: most measurement campaigns seem to focus on offsite measurements. You could specify the types of measurement campaigns that would help with the issues identified here (tanks and leaks)

Author: This is a great point, and we think it is worth highlighting in the discussion. We have added text acknowledging this in the later stages of the paper.

Manuscript L510: Given that locations of emissions sources from tanks are fewer (i.e., only possibilities are vents, PRVs, and thief hatches) compared to other equipment, site-level measurement campaigns (e.g., helicopter or airplane) could serve as more straight-forward alternatives to onsite measurement (which are particularly challenged for tanks that pose safety hazards and require access privileges). Such campaigns should be designed to refine the accuracy of the fraction and magnitude of unintentional emissions.

Line 242: one thing to consider though is that while the major equipment activity data are very recent, component count data is largely still from the 1990’s (both in GHGI and here)

Author: We have added text to this section to acknowledge this point.

Manuscript L239: The increase in estimated emissions from equipment leaks compared to the GHGI are due to our updated equipment-level emission factors; we know that the difference is not due to equipment-level activity factors because ours are nearly identical to the GHGI (see

Supplementary Methods 3). Equipment-level emission factors are themselves a function of both component-level emission data and component counts, and we acknowledge that our model relies heavily upon the same early 1990s data set as the GHGI for component counts.

Line 321: A large fraction of the equipment leaks data used in this study (if I'm correctly understanding, per table 1, 77% of the equipment leak components screened and also a large % of the measured leaks) are the same as used in the GHGI. Maybe there's a point to be made on the value of (and perhaps large impact of?) adding to the data set (so that it now includes HF wells, etc) in this study.

Author: We have added this caveat to the text.

Manuscript L354: It is finally worth noting that quantified emissions measurements (based on bagged measurements, and not those based on correlation equations) were included in this study's dataset. Although these measurements are small fraction (~7%) of our total dataset, the contribution is higher for specific components (**Supplementary Fig. 14**) emphasizing the importance of future data collection.

Author: We have also addressed some confusion in the interpretation of Table 1. Although the API [4] data set involved the most comprehensive screening of components (>100,000 components), only a fraction of these were quantified. The ERG [5] data set screened fewer components, but since all leaking components were quantified, this data set comprises a larger fraction of our emissions measurements. We can understand why this was confusing, so we have edited the table and added explanatory footnotes.

Manuscript L602:

Table 1: Summary of component-level datasets meeting inclusion criteria. Oil and gas methane emission measurement studies that reported raw data for quantified emissions measurements, fraction of components emitting, and component counts are summarized here. These studies are a subset of all studies that were examined closely, meeting inclusion criteria described. Detailed summary of each study's results are reported in Supplementary Methods 7.

Study ID	Location	Number of quantified leaks	Number of components screened	Leak volumes used	Component counts used	Components screened
Allen 2013 ³³	Various	646	NR	Y	N	Various components
Allen 2014 ⁴⁷	Various	378	378	Y	N	Pneum. controllers
Bell 2017 ⁶⁴	Fayetteville	247	NR	Y	N	Various components
ERG 2011 ³⁸	Barnett	1949	NR ¹	Y	N	Various components
Thoma 2017 ⁶⁵	Uintah	81	81	Y	N	Pneum. controllers
Pasci 2019 ³⁶	Various	192	54,618	Y	Y	Various components
API 1993 ³⁵	Various	251 ²	102,680	Y	Y	Various components
Clearstone 2018 ³⁷	Canada			N ³	Y	

NR = not reported

¹Screening counts are reported for several categories (connectors, valves, tanks) but counts are not comprehensive (see Supplementary Methods 4)

²Although only 251 data points from API 4598 were useful for quantification, 1780 leaking components were screened (i.e., only a subset of leaking components were quantified using the "bagging" technique)

³Given that leakage data was taken in Canada, we limit usage of this data to component counts

Line 518: Default EFs are "in some cases" based upon... The IPCC GL (both 2019 and 2006) use U.S. but also data from Australia, Canada, Norway, etc.

Author: The text has been edited.

Manuscript L532: Default emission factors for the petroleum and natural gas systems production-segment are in some cases based upon the same underlying data sets as the GHGI ⁶².

Supplementary Information:

Line 178: Recheck # of natural gas well sites used as input in calculating national emissions from Omara study (listed here as 498,000). It's probably lower. The total number of natural gas wells in the U.S. is around 500K, so if 498,000 well sites were assumed in the Omara study, that would mean around 1 well per site, when the average in the U.S. is around 2 wells per site (and the average in the Omara measured population is much higher).

Author: This was a typo. Although Omara et al. [13] refer to "natural gas production sites" in the title of their paper, the study assesses both oil and natural gas production sites in terms of the EPA definition (cutoff GOR = 100 mscf/bbl). The only difference between sites assessed in the Omara study and sites assessed in the Alvarez study is that Omara et al. exclude wells producing no natural gas. We have clarified the text.

Supplementary Information L164: The 498,000 US oil and natural gas production site assessed by Omara et al. (corresponding to approximately 813,615 wells, based on 2015 Enverus data) were binned into the same ten bins.

Line 280: Is the study EF for completions averaging out emissions over the course of a year so is a per day EF? GHGI EF is per event (so multiple days) so may want to be clearer on the comparison,

Author: The reviewer is correct. The activity data for completions and workovers is per event therefore the emission factor units are $\text{kg event}^{-1}\text{day}^{-1}$. The emission factors for the Greenhouse Gas Inventory were originally calculated with units per year (in the original document). These have been recalculated as $\text{kg event}^{-1}\text{day}^{-1}$, with the same units as the emission factors for our study. We have added a footnote to the table to make this clear.

Supplementary information L279: ³Activity units for completions are counts of events. Thus, the emission factor units are $\text{kg event}^{-1}\text{day}^{-1}$.

Line 481: Clarify which are leaker EFs versus population EFs

Author: We have added text to clarify this point.

Supplementary Information L477: Supplementary Table 5: Quantified measurement count and emissions factor (leaker) by study (16-fold scheme).

Line 657: Could also note info from Zimmerle et al 2019 on gathering and boosting (impact of measuring intermittent PCs for over 72 hours versus 15 mins)

Author: We have added a brief description of the Luck et al [14] study (the pneumatic component of Dr. Dan Zimmerle's 2019 gathering and boosting campaign [15]).

Supplementary Information L648: This concern regarding uncertainty introduced by intermittency has been noted recently by Luck et al. [14], who measured a smaller sample of pneumatic devices (72 devices) but for much longer periods (average of 76 hours, versus 15 minutes in Allen et al. [7]). Luck et al. caution the high degree of uncertainty in using short duration measurements to characterize emissions from pneumatic controllers and, especially for intermittent venting, "there is a low probability that the actuations occurring during that sample will be representative of the actuation rate over an extended period" [14].

Line 833: NSPS applies only to new or modified with a potential to emit of 6tpy VOC (so not all tanks)

Author: Thank you for the clarification. The text has been edited

Supplementary Information L819: US EPA NSPS subpart OOOO requires new or modified tanks (as of April 2011) with the potential to emit greater than 6 tons per year of VOC to install control devices such as flares or vapor recovery units designed to reduce volatile organic compound emissions by at least 95% ².

Line 1221: Flare stacks is only one category of flaring in GHGRP. I think this is described elsewhere in the SI but could also be made clear here.

Author: Supplementary Methods 6 describes how flaring emissions are reported in the GHGRP. We agree with the reviewer that it would be helpful to summarize these points in this section indicated by the reviewer. The following text has been added.

Supplementary Information L1134: Calculation of the GHGI emissions estimates based on GHGRP data is described in Supplementary Methods 6. Briefly, the operators report flaring activity and emissions to the GHGRP under the categories of well completions and workovers, routine venting of flash gas from storage tanks, flaring of associated gas, and other miscellaneous activities not covered under other categories. Although emissions are spread across various reporting tables, activity data is contained within the single table “EF_W_FLARE_STACKS_UNITS”. This table is useful for parameterizing activity data for our modified approach.

Line 1233: This isn't consistent with GHGRP data which does show flare stacks at gas wells

Author: The author thanks the reviewer for this clarification. We have checked the GHGRP data and using a similar approach to the EPA (which we describe in Supplementary Methods 6) we now partition flaring emissions between natural gas systems and petroleum systems. We describe this new approach in the text.

Supplementary Information L1149: Here, we use the same approach as the EPA to partition Subpart W data into natural gas and petroleum system categories (see Supplementary Methods 6, based on reported formation type by facility).

Supplementary Table 20: The total count of flare stacks in the United States oil and natural gas production segment.

	Natural gas systems	Petroleum systems
Total wells	433,430	571,761
Reported flaring wells ²	98,117	149,884
Reported wells ¹	290,710	214,993
Flaring wells per reported wells	0.3	0.7
Reported flare stacks ³	9,313	13,012
Reported flaring wells	98,117	149,884
Flare stacks per reported flaring wells	0.1	0.1
Total flare stacks	13,885	34,605

¹The number of wells reporting to "EF_W_FACILITY_OVERVIEW" is different from the number of wellheads reported to "EF_W_EQUIP_LEAKS_ONSHORE"

²Calculated by matching wells reporting to "EF_W_FACILITY_OVERVIEW" with facilities reporting to "EF_W_FLARE_STACKS_UNITS"

³Counted by summing unique flare stacks reported to "EF_W_FACILITY_OVERVIEW"

Further text is added later in the section (L1189).

These flare stack volumes are then binned into the same tranches used for well productivity volumes (**Supplementary Table 16-19**) to determine the fraction of wells flaring for each tranche. This parameter is treated like an equipment-level activity factor. For our two sets of natural gas and petroleum system tranches (**Supplementary Table 16-17** for natural gas systems and **Supplementary Table 18-19** for petroleum systems), flare stacks are assigned proportionally to well count.

Supplementary Table 22: Counts of flare stacks per well for oil wells with associated gas. This table corresponds to productivity tranches in **Supplementary Table 18**.

Bins (Mscf well ⁻¹ day ⁻¹)	Counts of wells	Count of flare stacks	Flare stacks per well
0-1	65606	5406	0.082
1-5	89163	5360	0.060
5-10	38948	2341	0.060
10-20	44410	2891	0.065
20-50	54343	2199	0.040
50-100	32775	1025	0.031
100-500	42848	616	0.014
500-1,000	7280	47	0.006
1,000-10,000	4464	24	0.005
10,000-inf	348	0	0.000

Supplementary Table 23: Counts of flare stacks per well for oil wells with no associated gas. This table corresponds to productivity tranches in **Supplementary Table 19**.

Bins (Mscf well ⁻¹ day ⁻¹)	Counts of wells	Count of flare stacks	Flare stacks per well
0-0.5	114,460	9432	0.082
0.5-1	27,149	2237	0.082
1-10	45,205	2717	0.060
10-inf	4,759	310	0.065

Supplementary Table 24: Counts of flare stacks per well for dry natural gas wells. This table corresponds to productivity tranches in **Supplementary Table 16**.

Bins (Mscf well ⁻¹ day ⁻¹)	Count	Counts of flare stacks	Flare stacks per well
0-1	38225	8326	0.218
1-5	65138	2975	0.046
5-10	43154	1426	0.033
10-20	44523	386	0.009
20-50	57148	154	0.003
50-100	27868	30	0.001
100-500	31429	20	0.001
500-1,000	6727	3	0.000
1,000-10,000	5875	1	0.000
10,000-inf	87	0	0.000

Supplementary Table 25: Counts of flare stacks per well for natural gas wells with associated oil production. This table corresponds to productivity tranches in **Supplementary Table 17**.

Bins_(Mscf well ⁻¹ day ⁻¹)	Count	Counts of flare stacks	Flare stacks per well
0-1	400	87	0.218
1-5	2880	132	0.046
5-10	3907	129	0.033
10-20	9384	81	0.009
20-50	30938	83	0.003
50-100	29916	32	0.001
100-500	30031	19	0.001
500-1,000	3094	1	0.000
1,000-10,000	2684	0	0.000
10,000-inf	22	0	0.000

Line 1241: Recheck—I believe the flare stack category only includes the flare stacks not reported elsewhere (e.g. tanks, completions)

Author: Although the GHGRP Table “EF_W_FLARE_STACKS_UNITS” only reports emissions for miscellaneous categories (i.e., those not reported in storage tanks or completions), it does include activity data for all categories. Only, in this case emissions are reported as 0. This is evident based on EPA instructions to reporters on the EPA GHGRP Confluence webpage. We have added a citation and additional explanation to the SI text in the later section where we elaborate on EPA GHGRP reporting.

SI L2331: Emissions listed in the table “EF_W_FLARE_STACKS_UNITS” are those that did not fit into one of the previously listed categories (flares from associated gas, flaring of vented gas during completion and workover events, and flares from storage tanks). However, according to EPA instructions to GHG reporters, “EF_W_FLARE_STACKS_UNITS” contains all activity data for all categories (on the EPA Confluence website it is stated “To the extent that monitoring information includes amounts associated with reported emission under [previously listed categories], those emissions should be deducted from the total emissions reported according to the methods in 98.233(n). If all emissions are deducted, then report 0 for the emissions and report all of the other required data elements”⁸⁴). This includes total volume of gas sent to flare, average GHG mole fraction, combustion efficiency, and fraction of gas sent to flare when it was unlit.

Additional fixes:

As a result of a careful inspection of our analysis, we discovered two minor bugs, both of which had a minor impact on the overall results.

The first was a mistake in calculating component counts for the 1993 American Petroleum Institute dataset [4]. This resulted in a less than 1% change in emissions for equipment leaks. We also fixed a

minor error we identified in Supplementary Fig. 13, which resulted in greater differentiation in mean emissions between dry gas producing sites and liquids producing sites. The error was in the figure plotting code and does not affect any other results in our analysis.

The combination of the above modifications and above reviewer requests required us to completely rerun our model. This resulted in an overall adjustment to our final result less than 1%. Thus, there was no overall change in our final conclusions.

Works cited

- [1] A. R. Brandt *et al.*, "Methane leaks from North American natural gas systems," *Science* (80-.), vol. 343, no. 6172, pp. 733–735, 2014.
- [2] Heath, G., Warner, E., Steinberg, D., Brandt, A.R., "Estimating U.S. Methane Emissions from the Natural Gas Supply Chain: Approaches, Uncertainties, Current Estimates, and Future Studies," Golden, CO, 2015.
- [3] Office of Inspector General EPA, "EPA Needs to Improve Air Emissions Data for the Oil and Natural Gas Production Sector," 2013.
- [4] Star Environmental, "Fugitive hydrocarbon emissions from oil and gas production operations. API Publication 4589," 1993.
- [5] (ERG) Eastern Research Group, "City of Fort Worth Natural Gas Air Quality Study," Morrisville, NC, 2011.
- [6] D. T. Allen *et al.*, "Measurements of methane emissions at natural gas production sites in the United States," *Proc. Natl. Acad. Sci. U. S. A.*, 2013, doi: 10.1073/pnas.1304880110.
- [7] D. T. Allen *et al.*, "Methane emissions from process equipment at natural gas production sites in the United States: Pneumatic controllers," *Environ. Sci. Technol.*, 2015, doi: 10.1021/es5040156.
- [8] C. S. Bell *et al.*, "Comparison of methane emission estimates from multiple measurement techniques at natural gas production pads," *Elementa*, 2017, doi: 10.1525/elementa.266.
- [9] E. D. Thoma, P. Deshmukh, R. Logan, M. Stovern, C. Dresser, and H. L. Brantley, "Assessment of Uinta Basin Oil and Natural Gas Well Pad Pneumatic Controller Emissions," *J. Environ. Prot. (Irvine, Calif.)*, 2017, doi: 10.4236/jep.2017.84029.
- [10] A. P. Pacsi *et al.*, "Equipment leak detection and quantification at 67 oil and gas sites in the Western United States," *Elementa*, 2019, doi: 10.1525/elementa.368.
- [11] Clearstone Engineering Ltd., "Update of Equipment, Component and Fugitive Emission Factors for Alberta Upstream Oil and Gas," Calgary, AB, 2018.
- [12] E. Buendia *et al.*, "Volume 2, Chapter 4: Fugitive Emissions - 2019 Refinement to the 2006 IPCC Guidelines for National Greenhouse Gas Inventories," Geneva, Switzerland, 2019.
- [13] M. Omara *et al.*, "Methane Emissions from Natural Gas Production Sites in the United States: Data Synthesis and National Estimate," *Environ. Sci. Technol.*, vol. 52, no. 21, pp. 12915–12925,

2018.

- [14] B. Luck *et al.*, "Multiday Measurements of Pneumatic Controller Emissions Reveal the Frequency of Abnormal Emissions Behavior at Natural Gas Gathering Stations," *Environ. Sci. Technol. Lett.*, 2019, doi: 10.1021/acs.estlett.9b00158.
- [15] D. Zimmerle *et al.*, "Characterization of Methane Emissions from Gathering Compressor Stations: Final Report," 2019.
- [16] Code of Federal Regulations, *Title 40 Part 60 Subpart OOOOa, Standards of Performance for Crude Oil and Natural Gas Facilities for which Construction, Modification or Reconstruction Commenced After September 18, 2015.* .
- [17] (EPA) Environmental Protection Agency, "Confluence: GHG Reporting Instructions: Subpart W Flares and Flare Stacks."
<https://ccdsupport.com/confluence/display/help/Subpart+W+Flares+and+Flare+Stacks>.